# Radial glia regulate vascular patterning around the developing spinal cord

Ryota L Matsuoka[1]*, Michele Marass[1], Avdesh Avdesh[1], Christian SM Helker[1], Hans-Martin Maischein[1], Ann S Grosse[2], Harmandeep Kaur[3], Nathan D Lawson[2], Wiebke Herzog[4,5], Didier YR Stainier[1]*

[1]Department of Developmental Genetics, Max Planck Institute for Heart and Lung Research, Bad Nauheim, Germany; [2]Department of Molecular, Cell and Cancer Biology, University of Massachusetts Medical School, Worcester, United States; [3]Department of Pharmacology, Max Planck Institute for Heart and Lung Research, Bad Nauheim, Germany; [4]Cells-in-Motion Cluster of Excellence, University of Muenster, Muenster, Germany; [5]Max Planck Institute for Molecular Biomedicine, Muenster, Germany

**Abstract** Vascular networks surrounding individual organs are important for their development, maintenance, and function; however, how these networks are assembled remains poorly understood. Here we show that CNS progenitors, referred to as radial glia, modulate vascular patterning around the spinal cord by acting as negative regulators. We found that radial glia ablation in zebrafish embryos leads to excessive sprouting of the trunk vessels around the spinal cord, and exclusively those of venous identity. Mechanistically, we determined that radial glia control this process via the Vegf decoy receptor sFlt1: *sflt1* mutants exhibit the venous over-sprouting observed in radial glia-ablated larvae, and sFlt1 overexpression rescues it. Genetic mosaic analyses show that sFlt1 function in trunk endothelial cells can limit their over-sprouting. Together, our findings identify CNS-resident progenitors as critical angiogenic regulators that determine the precise patterning of the vasculature around the spinal cord, providing novel insights into vascular network formation around developing organs.

*For correspondence: Ryota. Matsuoka@mpi-bn.mpg.de (RLM); Didier.Stainier@mpi-bn.mpg.de (DYRS)

## Introduction

Organ vascularization is a fundamental developmental event that is necessary for the establishment and maintenance of organ function. Previous work on the vascularization of different organs has shown that the process starts with angiogenic sprouting from the surrounding vasculature (*Kurz et al., 1996*; *Hogan et al., 2004*; *Sakaguchi et al., 2008*). For instance, the vascularization of the central nervous system (CNS) begins with angiogenic sprouting into neural tissues from the perineural vascular plexus (PNVP), a network that surrounds the CNS (*Feeney and Watterson, 1946*; *Strong, 1964*; *Kurz et al., 1996*; *Hogan et al., 2004*; *Bautch and James, 2009*). Similarly, liver vascularization has been shown to initiate by angiogenic sprouting from the vessels that first surround it (*Sakaguchi et al., 2008*). Although different organs undergo similar vascularization processes during development, it remains unknown how the surrounding vasculature, namely the arterial, venous, and lymphatic vessels, is assembled and patterned.

Prior studies have advanced our understanding of the cellular and molecular mechanisms governing CNS vascularization and have identified cells and factors that are important for CNS vascularization (*Haigh et al., 2003*; *Raab et al., 2004*; *Xu et al., 2004*; *Proctor et al., 2005*; *Stenman et al., 2008*; *Daneman et al., 2009*; *Ye et al., 2009*; *Junge et al., 2009*; *Kuhnert et al., 2010*; *Ma et al., 2013*; *Whiteus et al., 2014*; *Lacoste et al., 2014*; *Okabe et al., 2014*; *Zhou et al., 2014*;

*Arnold et al., 2014*; *Takahashi et al., 2015*; *Vanhollebeke et al., 2015*). Neuronal progenitors, including neuroepithelial cells and radial glia, play critical roles in the ingression of blood vessels into neural tissues (*Haigh et al., 2003*; *Raab et al., 2004*; *Stenman et al., 2008*; *Daneman et al., 2009*), as well as their stabilization in the cerebral cortex (*Ma et al., 2013*). Although it has been clearly shown that the ingression of blood vessels into neural tissues occurs through angiogenic sprouting from the PNVP in different organisms (*Kurz et al., 1996*; *Hogan et al., 2004*; *James et al., 2009*; *Takahashi et al., 2015*), and that the developing neural tube provides signals to recruit the endothelial cells (ECs) that form the PNVP (*Chapman and Papaioannou, 1998*; *Hogan et al., 2004*), how arterial and venous vessels are assembled around the CNS remains unclear. The terminals, or end-feet, of neuronal progenitors lie in close proximity to blood vessels in the basal lamina at the surface of the developing CNS (*Halfter et al., 2002*; *Kriegstein and Alvarez-Buylla, 2009*; *Lam et al., 2009*; *Vallon et al., 2014*; *Nakagawa et al., 2015*), raising the possibility that these cells are involved in regulating vascular development around the CNS. However, the role of the different CNS cell types in regulating the vascular architecture around the CNS remains to be investigated.

To systematically determine the extent to which different CNS cell types control vascular development around the CNS, we utilized the zebrafish larva as a model system as its optical clarity allows the easy visualization of its stereotyped yet simple vascular pattern (*Isogai et al., 2001*). In addition, the use of an established genetic cell ablation method that utilizes the bacterial enzyme Nitroreductase (NTR) to induce apoptotic cell death upon administration of its substrate prodrug metronidazole (Mtz) (*Curado et al., 2007*; *Pisharath et al., 2007*; *Curado et al., 2008*) allowed us to ablate different CNS cell types in a spatially and temporally controlled manner. In combination with genetic mutants and pharmacological tools, we used this ablation technology to analyze the roles of distinct CNS cell types in patterning the vasculature around the spinal cord.

## Results

### Physical proximity between the end-feet of spinal cord radial glia and extraneural vessels during development

As the developing zebrafish CNS is known to be composed of different glial and neuronal cell types, we first categorized them into major cell type groups: radial glia neuronal progenitors, neurons, oligodendrocytes, and microglia. Unlike in the mammalian CNS, radial glia neuronal progenitors persist into adulthood in many regions of the zebrafish CNS (*Lam et al., 2009*; *Than-Trong and Bally-Cuif, 2015*); and to date, there is no clear evidence for bona fide astrocytes in the developing zebrafish CNS at embryonic or early larval periods, the stages we analyzed in this study.

Prior to ablating these different cell types, we first conducted an extensive analysis of potential interaction sites where the vasculature and cell bodies and/or processes from these CNS cell types are physically adjacent at different developmental stages. Through these analyses, we identified that cellular processes marked by *TgBAC(gfap:*Gfap-EGFP*)$^{zf167}$* expression lie in close proximity to different vessels in the trunk, including intersegmental vessels (ISVs) and the vertebral arteries (VTAs) (*Figure 1A–C'*). *TgBAC(gfap:*Gfap-EGFP*)* expression prominently and almost exclusively marks one cell type in the developing spinal cord, one that extends long radial processes from the ventricular lumen towards the pial surface, a morphological feature that is characteristic of radial glia (*Bernardos and Raymond, 2006*; *Lam et al., 2009*; *Than-Trong and Bally-Cuif, 2015*) (*Figure 1B–C'* and *Video 1*). Through careful analyses at different developmental stages, we also observed that the physical proximity between radial glia end-feet and ISVs as well as VTAs persists throughout development. These observations along with the fact that a similar arrangement can be seen in the developing mouse trunk (*Vallon et al., 2014*) motivated us to investigate the potential role of spinal cord radial glia in the development of these trunk vessels.

We thus started by ablating radial glia during development. In parallel, we also ablated neurons, since the role for neurons in vascular development around the CNS had not been systematically addressed. To ablate radial glia, we generated a new transgenic (Tg) line that drives Gal4ff under the control of the same *gfap* BAC promoter used for *TgBAC(gfap:gfap-EGFP)$^{zf167}$* (*Lam et al., 2009*). We confirmed that this newly generated *TgBAC(gfap:gal4ff)$^{s995}$* line indeed marks radial glia in the developing spinal cord by crossing it with the previously characterized *Tg(gfap:GFP)$^{mi2001}$* line (*Bernardos and Raymond, 2006*), as well as by performing immunohistochemistry using a pan-

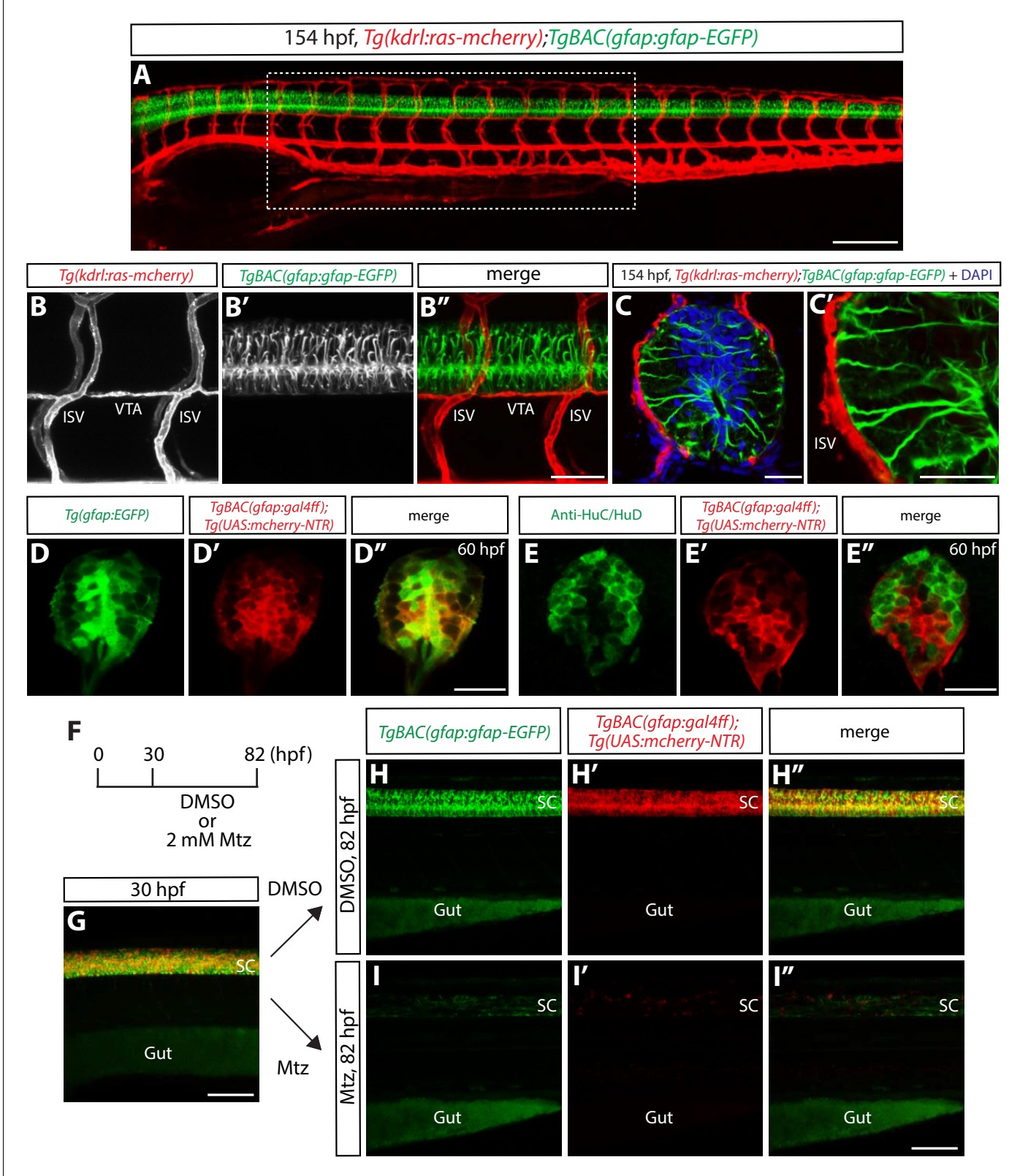

**Figure 1.** Physical proximity between the end-feet of spinal cord radial glia and extraneural vessels during development. (**A**) Lateral view of a 154 hpf *Tg(kdrl:ras-mcherry);TgBAC(gfap:gfap-EGFP)* trunk. The region inside the dashed line corresponds to the 10 somites where all analyses and quantifications were performed. Scale bar, 200 µm. (**B–B″**) High magnification images of a *Tg(kdrl:ras-mcherry);TgBAC(gfap:gfap-EGFP)* trunk at 154 hpf. End-feet of radial glia and ISVs or VTAs are physically adjacent. ISVs: intersegmental vessels, VTAs: vertebral arteries. Scale bar, 100 µm. (**C–C′**)
*Figure 1 continued on next page*

*Figure 1 continued*

High magnification confocal single-plane images of 154 hpf *Tg(kdrl:ras-mcherry);TgBAC(gfap:gfap-EGFP)* trunk section counterstained with DAPI. Radial glia end-feet lie in close proximity to ISVs (**C'**). Scale bars, 20 μm. (**D–D"**) 60 hpf *Tg(gfap:EGFP);TgBAC(gfap:gal4ff);Tg(UAS:mcherry-NTR)* trunk spinal cord section. EGFP⁺ radial glia and mCherry⁺ cells are largely co-localized. Scale bar, 20 μm. (**E–E"**) 60 hpf *TgBAC(gfap:gal4ff);Tg(UAS:mcherry-NTR)* trunk spinal cord section immunostained for HuC/HuD (green). EGFP⁺ neurons and mCherry⁺ cells are largely segregated. Scale bar, 20 μm. (**F**) Time course of nitroreductase (NTR)/metronidazole (Mtz)-mediated cell ablation of radial glia for the panels (**G–I"**). (**G**) 30 hpf *TgBAC(gfap:gfap-EGFP); TgBAC(gfap:gal4ff);Tg(UAS:mcherry-NTR)* trunk. *TgBAC(gfap:Gfap-EGFP)* expression and *TgBAC(gfap:gal4ff);Tg(UAS:mCherry-NTR)* expression are observed in the spinal cord in an overlapping manner. SC: spinal cord. Scale bar, 100 μm. (**H–H"** and **I–I"**) 82 hpf *TgBAC(gfap:gfap-EGFP);TgBAC(gfap: gal4ff);Tg(UAS:mcherry-NTR)* trunk after treatment with DMSO (**H–H"**) or 2 mM Mtz (**I–I"**) starting at 30 hpf. Unlike DMSO-treated fish that show strong co-expression of *TgBAC(gfap:Gfap-EGFP)* and *TgBAC(gfap:gal4ff);Tg(UAS:mCherry-NTR)* in their spinal cord, Mtz-treated fish show a dramatic reduction of this co-expression. Scale bar, 100 μm.

The following figure supplement is available for figure 1:

**Figure supplement 1.** Characterization of radial glia or neuronal ablation by the NTR/Mtz-mediated cell ablation method.

neuronal anti-HuC/HuD antibody (*Figure 1D–E"*). We then crossed *TgBAC(gfap:gal4ff)^s995* fish to *Tg (UAS-E1b:Eco.NfsB-mCherry)^c264* fish, which carry a *NTR-mcherry* fusion gene downstream of UAS (upstream activating sequence), in order to generate *TgBAC(gfap:gal4ff)^s995;Tg(UAS-E1b:Eco.NfsB-mCherry)^c264* animals, hereafter abbreviated *Tg(gfap:NTR)*. To ablate neurons, we used *Tg(elavl3: gal4-vp16)^psi1;Tg(UAS-E1b:Eco.NfsB-mCherry)^c264* fish, hereafter abbreviated *Tg(elavl3:NTR)*. We first tested and confirmed that these lines can induce radial glia or neuron ablation robustly upon administration of Mtz (*Figure 1F–I"* and *Figure 1—figure supplement 1C–I"*). Furthermore, we confirmed by acridine orange staining and TUNEL assays that these lines induce apoptotic cell death specifically in the CNS (data not shown), and that when we initiated cell ablation at 30 hpf, radial glia and neurons in the spinal cord are largely segregated as determined by the expression of fluorescent proteins driven by the *gfap* (*Lam et al., 2009*) and *elavl3* (*Park et al., 2000*) promoters, respectively (*Figure 1—figure supplement 1A and B*).

## Genetic ablation of CNS radial glia during development leads to disrupted patterns of the vasculature around the spinal cord

We then investigated whether and how radial glia or neurons regulate vascular development around the CNS by crossing *Tg(gfap:NTR)* or *Tg(elavl3:NTR)* fish into the EC reporter Tg line, *Tg(kdrl: EGFP)^s843*. To analyze the role of each cell type in vascular development, we performed the genetic ablation at early developmental stages. We set up an experiment where dechorionated embryos were treated with 2 mM Mtz starting at 30 hpf and assessed vascular patterning at 154 hpf (*Figure 2A*). We observed that radial glia-ablated fish exhibited largely comparable patterns of the trunk vasculature to controls; however, we found a severe vascular patterning defect in a specific region of radial glia-ablated trunks (*Figure 2B and C*; quantification shown in *Figure 2F*; n = 25 fish for each group). Namely, radial glia-ablated fish developed ectopically sprouting vessels between the ISVs in the dorsal part of the trunk adjacent to the spinal cord (*Figure 2C* and *Video 2*). These ectopic vessels were observed at the surface of, but not inside, the spinal cord (*Video 2*) and were not observed in control DMSO or 2 mM Mtz treated fish (*Figure 2B*; data not shown). Intriguingly, we also did not observe a similar vascular defect in neuron-ablated fish (*Figure 2D and E*; n = 25), suggesting a

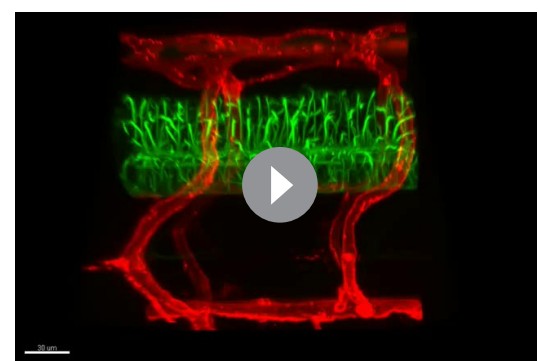

**Video 1.** *TgBAC(gfap:gfap-EGFP);Tg(kdrl:ras-mcherry)* trunk at 132 hpf. A vast majority of the cells that are marked by *TgBAC(gfap:Gfap-EGFP)* expression appear to have long radial processes towards the surface of the spinal cord, a morphology characteristic of radial glia. Scale bar, 20 μm.

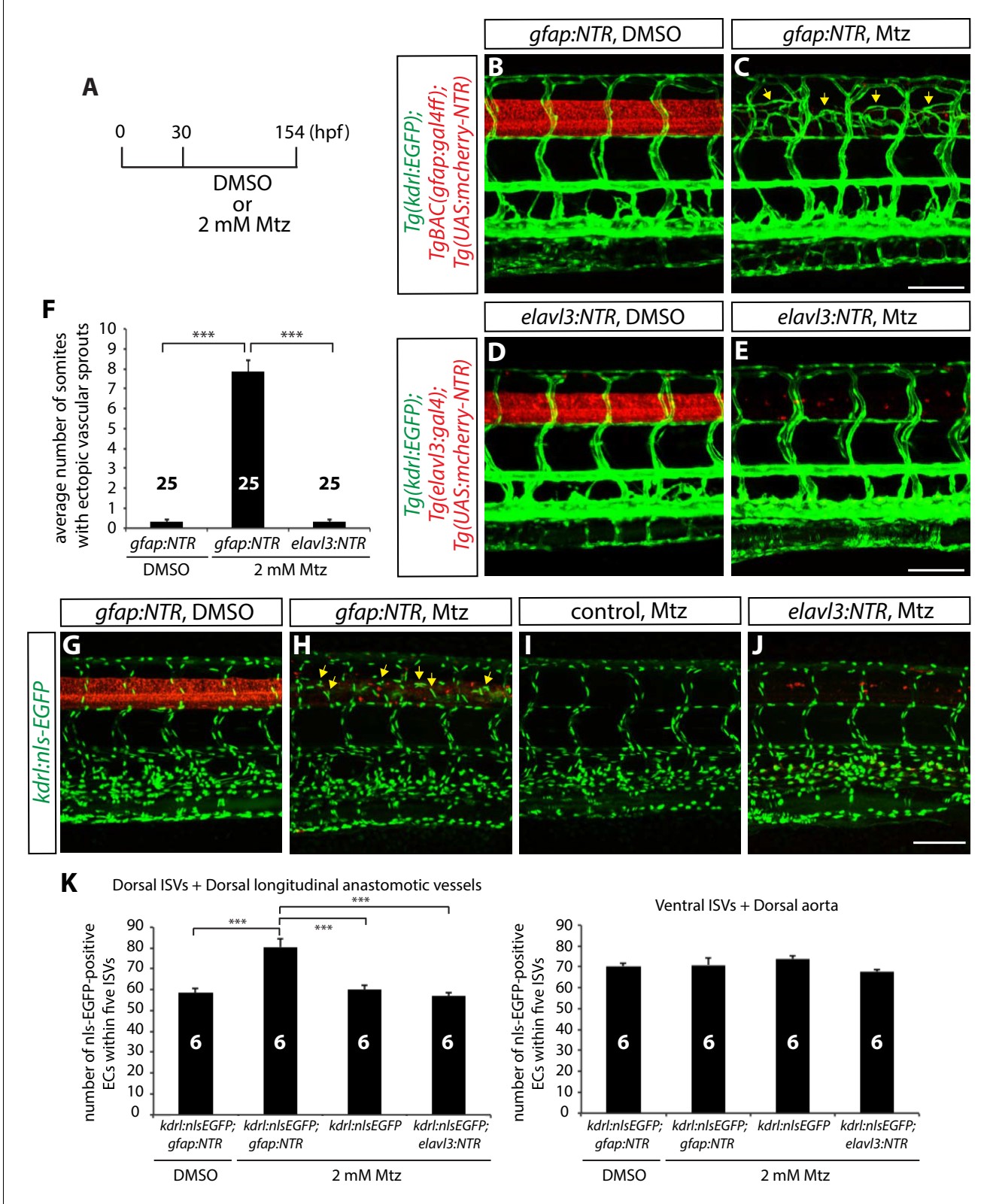

**Figure 2.** Genetic ablation of CNS radial glia in embryos leads to excessive sprouting and an increased number of dorsal trunk vascular ECs around the spinal cord. (A) Experimental time course for the panels (B–F). (B–E) 154 hpf *TgBAC(gfap:gal4ff);Tg(UAS:mcherry-NTR);Tg(kdrl:EGFP)* (B and C) and *Tg(elavl3:gal4);Tg(UAS:mcherry-NTR);Tg(kdrl:EGFP)* (D and E) trunks after treatment with DMSO (B and D) or 2 mM Mtz (C and E) between 30 and 154 hpf. Genetic ablation of radial glia, but not of neurons, leads to ectopic vessel sprouting in the dorsal part of the trunk (yellow arrows, C). Scale bars, 100

*Figure 2 continued on next page*

*Figure 2 continued*

µm. (F) Quantification of average number of somites that showed ectopic blood vessels (10 somites examined per animal; 25 animals examined per condition). This quantification was performed at 154 hpf; fish after radial glia ablation show a dramatic increase in the number of somites with ectopic blood vessels. (G–J) 154 hpf *TgBAC(gfap:gal4ff);Tg(UAS:mcherry-NTR);Tg(kdrl:nls-EGFP)* (G and H), *Tg(kdrl:nls-EGFP)* (I) and *Tg(elavl3:gal4);Tg(UAS:mcherry-NTR);Tg(kdrl:nls-EGFP)* (J) trunks after treatment with DMSO (G) or 2 mM Mtz (H–J) between 30 and 154 hpf. Radial glia-ablated fish show ectopic ECs in the dorsal part of the trunk (yellow arrows, H). Scale bar, 100 µm. (K) Quantification of nls-EGFP$^+$ EC number within 5 ISVs (6 animals examined per condition). Radial glia ablation leads to an approximately 30% increase in EC number in the dorsal part of the trunk (left panel), whereas no significant differences in EC number were observed in the ventral part of the trunk (right panel). See also *Figure 2—source data 1* for quantification. In all panels, values represent means ± SEM (*** indicates p<0.001 by one-way analysis of variance (ANOVA) followed by Tukey's HSD test).

The following source data and figure supplement are available for figure 2:

**Source data 1.** Quantification of EC number in the dorsal and ventral part of trunks.

**Figure supplement 1.** Microglia, macrophages, pericytes, and/or neutrophils do not modulate trunk vascular patterning.

specific role for radial glia in regulating this aspect of trunk vascular patterning. Importantly, these observations also show that the cell responses evoked by massively-induced apoptotic cell death in the CNS are not sufficient to cause this vascular phenotype (*Figure 2B–F*). Thus, radial glia appear to function as negative angiogenic regulators during the patterning of dorsal ISVs around the spinal cord during development.

It is important to note that the vascular phenotypes after radial glia ablation are unlikely to be caused by the reduction or loss of other CNS or PNS cell types such as oligodendrocytes, microglia, or Schwann cells, since animals treated with cyclopamine, which leads to a dramatic loss of spinal cord oligodendrocytes and oligodendrocyte precursor cells, *irf8*$^{st96}$ mutants, which lack microglia (*Shiau et al., 2015*), or *erbb2*$^{st61}$ mutants, which have a reduced number of Schwann cells (*Lyons et al., 2005*), did not display these vascular phenotypes (*Figure 3A–H*). Furthermore, cell types such as macrophages, neutrophils, or pericytes, which play roles in angiogenesis and/or vascular remodeling (*Gong and Koh, 2010*; *Ribatti et al., 2011*; *Newman and Hughes, 2012*; *Arnold and Betsholtz, 2013*), are also unlikely to be involved; the use of mutants, including *irf8*$^{st96}$, which lack microglia and have a dramatically reduced number of macrophages up to 5–6 dpf (*Shiau et al., 2015*), and *pdgfrb*$^{um148}$, which have a reduced number of pericytes (*Kok et al., 2015*; *Ando et al., 2016*), as well as an ablation Tg line, *Tg(mpx:gal4):Tg(UAS-E1b:Eco.NfsB-mCherry)*, which leads to a dramatically reduced number of neutrophils following cell ablation (data not shown), did not affect the severity of the vascular phenotypes after radial glia ablation (*Figure 2—figure supplement 1*; data not shown). Thus, these findings strongly suggest that specifically radial glia, but not other CNS-resident cell types including those derived from radial glia, regulate this vascular patterning process.

## Genetic ablation of radial glia leads to excessive sprouting and an increased number of dorsal trunk ECs

Identification of a specific role played by radial glia in trunk vascular patterning led us to further characterize the phenotype. We first determined whether the ectopic dorsal trunk vessels after radial glia ablation were caused by mis-migration or an increased number of ECs. For this purpose, we quantified the number of ECs that constitute the trunk vasculature using

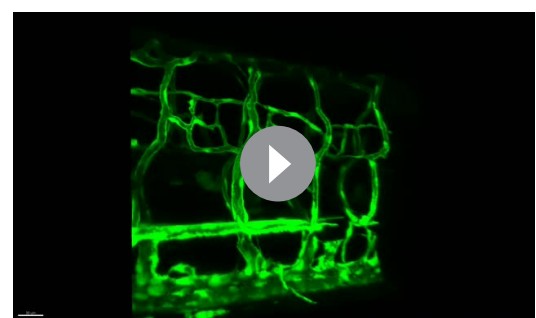

**Video 2.** Ectopic vessels after radial glia ablation lie at the surface of, but not inside, the spinal cord. 7 dpf *TgBAC(gfap:gal4ff);Tg(UAS:mcherry-NTR);Tg(kdrl:EGFP)* trunk vasculature visualized by *Tg(kdrl:EGFP)* expression. Ectopic vessels after radial glia ablation lie at the surface of, but not inside, the spinal cord. Scale bar, 30 µm.

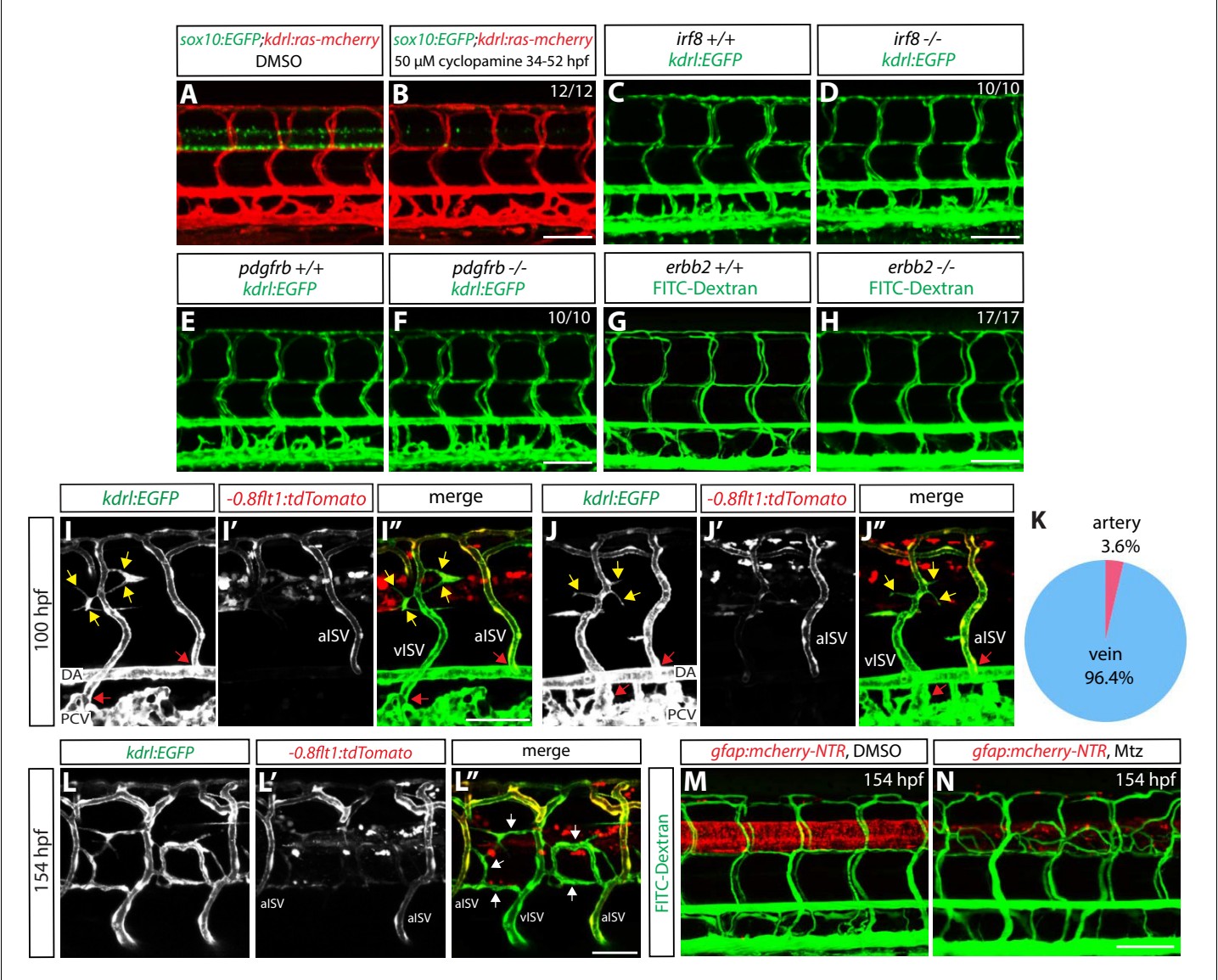

**Figure 3.** Genetic ablation of CNS radial glia leads to selective over-sprouting of venous ISVs, and ablation of other CNS or PNS cell types does not cause this phenotype. (A and B) 154 hpf *Tg(sox10:EGFP);Tg(kdrl:ras-mcherry)* trunks after treatment with DMSO (A) or 50 µM cyclopamine (B) between 34 and 52 hpf. The ectopic ISV sprouting phenotype observed after radial glia ablation is not found in fish treated with cyclopamine, which show a dramatically reduced number of spinal cord oligodendrocytes and oligodendrocyte precursor cells. Scale bar, 100 µm. (C and D) 154 hpf *Tg(kdrl:EGFP)* *irf8*[+/+] (C) and *irf8*[-/-] (D) trunk vasculature. The ectopic ISV sprouting phenotype observed after radial glia ablation is not seen in *irf8*[-/-] fish. Scale bar, 100 µm. (E and F) 154 hpf *Tg(kdrl:EGFP)* *pdgfrb*[+/+] (E) and *pdgfrb*[-/-] (F) trunk vasculature. The ectopic ISV sprouting phenotype observed after radial glia ablation is not seen in *pdgfrb*[-/-] fish. Scale bar, 100 µm. (G and H) 154 hpf *erbb2*[+/+] (G) and *erbb2*[-/-] (H) trunk vasculature visualized by FITC-dextran microangiography. The ectopic ISV sprouting phenotype observed after radial glia ablation is not seen in *erbb2*[-/-] fish. Scale bar, 100 µm. (I–I" and J–J") 100 hpf *TgBAC(gfap:gal4ff);Tg(UAS:mcherry-NTR);Tg(kdrl:EGFP);Tg(-0.8flt1:tdTomato)* fish that were treated with 2 mM Mtz starting at 30 hpf. *Tg(-0.8flt1:tdTomato)* expression labels arterial ISVs strongly and venous ISVs weakly (I' and J'), and ectopic vessel sprouts after radial glia ablation derive from venous ISVs in most cases (yellow arrows). Red arrows point to ISVs' connection sites with their axial vessels, namely the dorsal aorta (DA) and posterior cardinal vein (PCV), confirming the identity of aISVs and vISVs as revealed by the differential *Tg(-0.8flt1:tdTomato)* expression in these vessels. Scale bar, 50 µm. (K) Quantification after radial glia ablation of ectopic sprouts that derive from aISVs or vISVs at 100 hpf. A vast majority of the ectopic sprouts derive from vISVs (106 out of 110 ectopic sprouts derived from vISVs in 17 fish). (L–L") High magnification images of a 154 hpf *TgBAC(gfap: gal4ff);Tg(UAS:mcherry-NTR);Tg(kdrl:EGFP);Tg(-0.8flt1:tdTomato)* trunk after radial glia ablation. The animal was treated with 2 mM Mtz between 30 and 154 hpf. Ectopic vessels in the dorsal trunk after radial glia ablation exhibit weak *Tg(-0.8flt1:tdTomato)* expression as vISVs do (white arrows, L"). Scale bar, 50 µm. (M and N) 154 hpf *TgBAC(gfap:gal4ff);Tg(UAS:mcherry-NTR)* larvae that were injected with FITC-Dextran nanocrystals. Animals were treated with DMSO (M) or 2 mM Mtz (N) between 30 and 154 hpf, and then injected with FITC-Dextran nanocrystals at 154 hpf. FITC-dextran nanocrystals that were injected into the common cardinal vein circulated and labeled the trunk vasculature as shown in the DMSO-treated larva (M). Ectopic blood

*Figure 3 continued on next page*

*Figure 3 continued*
vessels that emerged after radial glia ablation were labelled by FITC-dextran nanocrystals (N), showing that these ectopic vessels are part of the circulatory loop. Scale bar, 100 μm.
The following figure supplements are available for figure 3:

**Figure supplement 1.** The time course of ectopic blood vessel growth after radial glia ablation, and the emergence of ectopic blood vessels in the trunk regions where radial glia are robustly ablated.

**Figure supplement 2.** Characterization of *Tg(-0.8flt1*:tdTomato) expression during development, and validation of *Tg(hsp70l:sflt1, cryaa:cerulean)* and *Tg(hsp70l:sflt4, cryaa:cerulean)* lines.

a *Tg(kdrl:NLS-EGFP)^{ubs1}* reporter line. We divided the trunk vasculature into dorsal and ventral parts for this quantification. We counted the number of ECs in the dorsal part, including the dorsal ISVs, the VTAs, and the dorsal longitudinal anastomotic vessels (DLAVs) and found an approximately 30% increase in EC numbers in radial glia-ablated fish compared to control or neuron-ablated fish (*Figure 2G–K* and *Figure 2—source data 1*; n = 6 for each group). In contrast, the number of ECs in the ventral part, where we counted ECs in ventral ISVs and the dorsal aorta (DA), did not differ between radial glia-ablated, neuron-ablated, and control fish (*Figure 2G–K* and *Figure 2—source data 1*; n = 6 for each group). These results show that radial glia ablation leads to ectopic sprouting and an increased number of ECs specifically in the dorsal trunk. Thus, these cell ablation experiments identify a unique role for CNS radial glia in regulating trunk vascular patterning around the spinal cord.

## Radial glia ablation leads to selective over-sprouting of venous ISVs

To gain mechanistic insight into how radial glia regulate trunk vascular patterning, we first characterized in detail the ectopic vessels that emerge after radial glia ablation. We analyzed the growth progression of the ectopic sprouts at different developmental stages of radial glia-ablated fish that were treated with Mtz between 30 and 54 hpf. Around 74 hpf, when a majority of radial glia were ablated, we observed ectopic vessel sprouting, and these ectopic vessels continued to grow until 6–7 dpf (*Figure 3—figure supplement 1A–P*; data not shown). Furthermore, by comparing radial glia-ablated fish that were treated with Mtz for different lengths of time, we found that the emergence of the ectopic vessels correlated with regions where radial glia were robustly ablated (*Figure 3—figure supplement 1Q–W'*), indicating that radial glia modulate nearby vessels to prevent their over-sprouting. To determine when radial glia regulate vascular patterning in the trunk, we ablated radial glia at different developmental stages. Ectopic vessel sprouting emerged only when we initiated radial glia ablation at early developmental stages, i.e., before 60 hpf, suggesting a critical developmental time window of negative angiogenic action of radial glia and/or the vessels' response (data not shown).

We next assessed whether the ectopic vessels that emerged after radial glia ablation were arterial or venous using the *Tg(-0.8flt1:tdTomato)^{hu5333}* line, which has been reported to mark arterial ISVs strongly and venous ISVs weakly (*Bussmann et al., 2010*). The *Tg(-0.8flt1:tdTomato)* line indeed labeled arterial and venous ISVs differently in our hands as well, and this differential expression of tdTomato in aISVs versus vISVs became clear at 56 hpf and even more obvious at 80 hpf (*Figure 3—figure supplement 2A–C''*). By analyzing ectopic sprouts after radial glia ablation in 100 hpf *Tg(-0.8flt1:tdTomato);Tg(kdrl:EGFP);Tg(gfap:NTR)* fish, we found that a vast majority of the ectopic sprouts derived from vISVs (*Figure 3I–K*; 106 out of 110 sprouts derived from vISVs; n = 17 fish). In addition to this assessment by the differential tdTomato expression in aISVs versus vISVs, we also carefully examined ISVs' connections with their axial vessels, namely the DA and posterior cardinal vein (PCV), and observed that the ectopic sprouts after radial glia ablation emerge selectively from vISVs as they were directly connected to the PCV (*Figure 3I–J''*). At 154 hpf, when the ectopic sprouts deriving from vISVs appeared to be more mature and integrate within the circulatory loop (*Figure 3M and N*), a majority of these ectopic vessels showed weak *Tg(-0.8flt1:*tdTomato) expression similar to that of vISVs (*Figure 3L–L''*), suggesting that these vessels are of venous identity.

To further test whether this effect of radial glia ablation is selective for vISVs, we performed experiments that genetically reduce the number of vISVs. If the effect of radial glia ablation is indeed specific for vISVs, reduction of their numbers should lead to a decrease in excessive vessel sprouting. To achieve this vISV reduction, we generated a *Tg(hsp70l:sflt4, cryaa:cerulean)* line, hereafter abbreviated *Tg(hsp70l:sflt4)*, which upon heat shock overexpresses a soluble form of Flt4/Vegfr3 (sFlt4), a decoy receptor for Vegfc and Vegfd (*Ruiz de Almodovar et al., 2009*). This transgene also contains the eye-marker cassette, *cryaa:cerulean* (*Kurita et al., 2003*; *Hesselson et al., 2009*), to facilitate the identification of transgenic animals. Compromised Vegfc/Vegfr3 signaling was previously shown to cause a dramatically reduced number of vISVs as assessed in zebrafish *vegfc* and *vegfr3* mutants (*Hogan et al., 2009b*; *Le Guen et al., 2014*). We found that overexpression of sFlt4 by heat shocking the embryos at 29, 36, and 43 hpf, stages when angiogenic sprouting from the PCV actively occurs, recapitulated these mutant phenotypes, resulting in a reduced number of vISVs (approximately 9% vISVs remaining per transgenic fish on average, *Figure 4A,C,E*, and *Figure 4—source data 1*; n = 23). Notably, when we performed radial glia ablation in these fish with a reduced number of vISVs, we found that the ectopic sprouting after radial glia ablation was dramatically reduced as compared to non-heat shocked or heat shocked control siblings (*Figures 4B–D and F*; n = 15 for each group), which have approximately 55% vISVs on average (*Figure 4E* and *Figure 4—source data 1*; n = 18). These results provide further evidence that radial glia ablation causes the excessive sprouting of venous ECs predominantly, and thus suggest that radial glia negatively regulate the over-sprouting of dorsal trunk venous vessels during development.

## Spinal cord radial glia negatively modulate Vegf/Vegfr2 signaling to prevent ectopic sprouting of venous ISVs

To determine which angiogenic signal(s) triggers ectopic vessel sprouting after radial glia ablation, we used pharmacological and genetic tools that inhibit different angiogenic signaling pathways. We first used small molecule inhibitors to identify signaling pathways responsible for the ectopic vessel sprouting. Specifically, we treated embryos which were pre-incubated with 2 mM Mtz between 30 and 54 hpf, with the following chemicals from 54 until 130 hpf: 2 μM sunitinib, a broad receptor tyrosine kinase inhibitor, 2.5 μM SKLB1002, a selective Vegfr2 inhibitor, 10 μM LY294002, a selective PI3 kinase inhibitor, 5 μM DMHI, a selective Bmp receptor inhibitor, 15 μM AG1295, a selective Pdgf receptor inhibitor, and 3 μM PD158780, a selective ErbB receptor tyrosine kinase inhibitor. Among the chemicals tested, selective inhibitors for Bmp, Pdgf, or ErbB receptors did not block ectopic vessel sprouting after radial glia ablation (*Figure 4K*; n = 13–17 for each treatment group), suggesting that these signaling pathways are not required for inducing the ectopic vessel sprouting. In contrast, inhibitors for receptor tyrosine kinases (sunitinib), Vegfr2 (SKLB1002), and PI3 kinase (LY294002) significantly blocked ectopic vessel sprouting, indicating that the Vegfr2 signaling pathway is responsible for the phenotype observed after radial glia ablation (*Figure 4G–K*; n = 16 for each treatment group).

To further test these hypotheses, we also used genetic tools that block the Pdgf, Bmp, or Vegf signaling pathways. Consistent with the chemical treatment results, neither the *pdgfrb[um148]* mutants nor overexpression of the BMP inhibitor Noggin3 using *Tg(hsp70l:nog3)[fr14]* fish (*Chocron et al., 2007*), blocked the ectopic vessel sprouting after radial glia ablation (*Figure 2—figure supplement 1E–H* and *Figure 4—figure supplement 1A–E*; n = 11 for *pdgfrb[-/-]* fish and n = 24 for *Tg(hsp70l: nog3)* fish with multiple heat shocks). To investigate which Vegf ligand(s) and receptor(s) are responsible for inducing the ectopic vessel sprouting after radial glia ablation, we generated and used a *Tg (hsp70l:sflt1, cryaa:cerulean)* line, hereafter abbreviated *Tg(hsp70l:sflt1)*, which upon heat shock overexpresses a soluble form of Flt1/Vegfr1 (sFlt1), a decoy receptor for Vegfa, Vegfb, and PlGF (Placental growth factor) (*Ruiz de Almodovar et al., 2009*), as well as the *Tg(hsp70l:sflt4)* line as described earlier (*Figure 3—figure supplement 2D–S*). We found that ectopic vessel sprouting after radial glia ablation was dramatically blocked in *Tg(hsp70l:sflt1)* but not *Tg(hsp70l:sflt4)* fish, after heat shocking the animals every 12 hr starting at 62 hpf, a stage prior to the onset of ectopic vessel sprouting (*Figure 4—figure supplement 1F–H*). Therefore, these results indicate that ectopic vessel sprouting after radial glia ablation is dependent on Vegfr2, but not Vegfr3, signaling. Consistent with the chemical treatment results, these findings identify Vegf/Vegfr2 signaling as a critical signaling pathway responsible for ectopic vessel sprouting after radial glia ablation, indicating that radial glia negatively modulate this signaling pathway during trunk vasculature development.

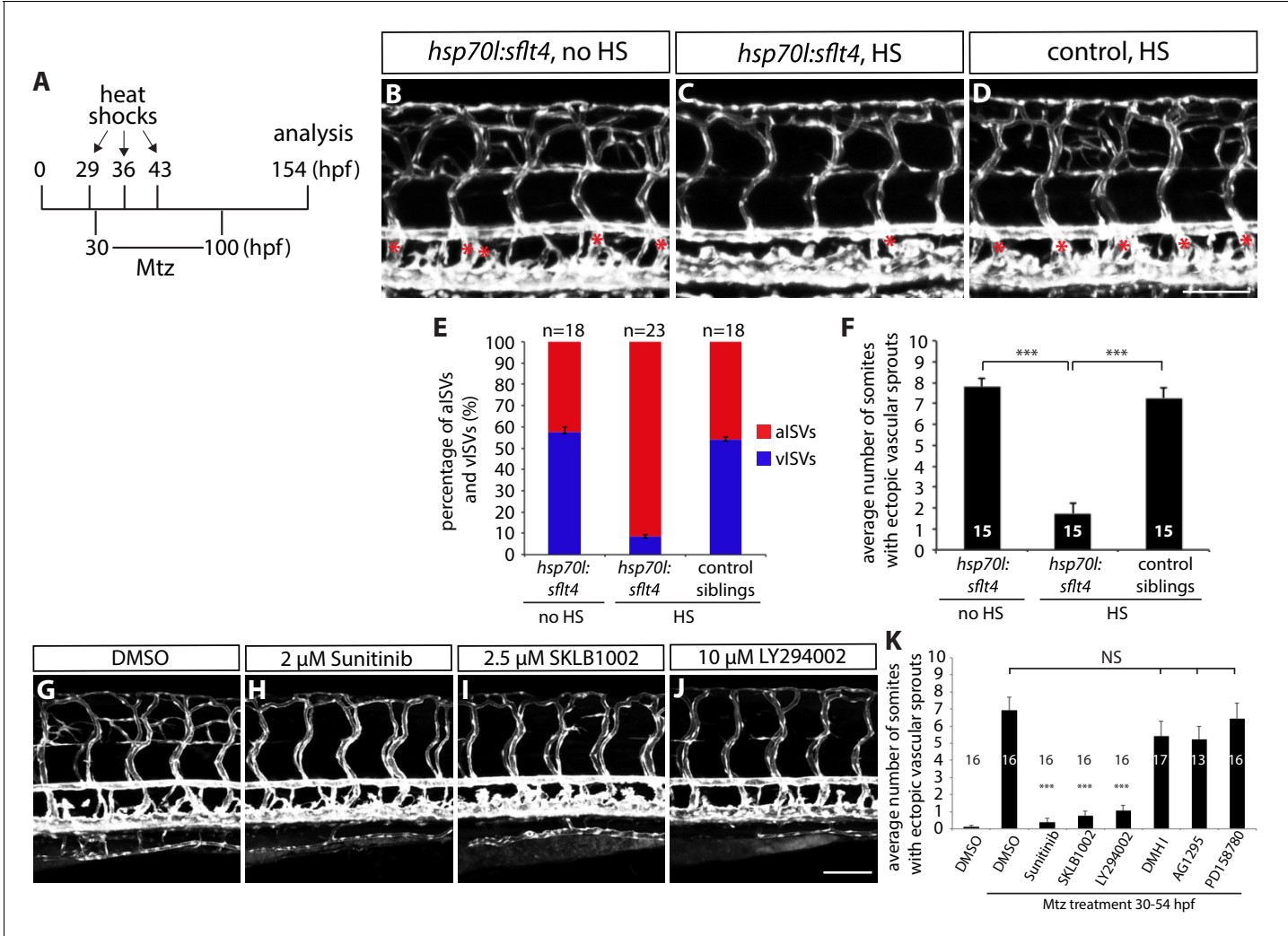

**Figure 4.** Fish with reduced numbers of venous ISVs exhibit dramatically fewer ectopic sprouts after radial glia ablation. (A) Experimental time course for data shown in (B–F). (B–D) 154 hpf *TgBAC(gfap:gal4ff);Tg(UAS:mcherry-NTR);Tg(kdrl:EGFP);Tg(hsp70l:sflt4, cryaa:cerulean)* (B and C) and *TgBAC (gfap:gal4ff);Tg(UAS:mcherry-NTR);Tg(kdrl:EGFP)* (D) trunks after treatment with 2 mM Mtz between 30 and 100 hpf. Animals were subject to no heat shock (B) or multiple heat shocks at 29, 36, and 43 hpf (C and D). Red asterisks indicate vISVs, and inhibition of Vegfc/Vegfr3 signaling by overexpression of a soluble form of Flt4/Vegfr3 (sFlt4) dramatically reduced the number of vISVs (B–D). Under these conditions, significantly reduced ectopic ISV sprouting was observed after radial glia ablation (C). HS: heat shock. Scale bar, 100 µm. (E) Percentage of aISVs and vISVs in 154 hpf *TgBAC (gfap:gal4ff);Tg(UAS:mcherry-NTR);Tg(kdrl:EGFP);Tg(hsp70l:sflt4, cryaa:cerulean)* and *TgBAC(gfap:gal4ff);Tg(UAS:mcherry-NTR);Tg(kdrl:EGFP)* larvae that were given no heat shock or multiple heat shocks at 29, 36, and 43 hpf. See also *Figure 4—source data 1* for quantification. (F) Quantification of average number of somites that showed ectopic blood vessels (10 somites examined per animal; 15 animals examined per condition). Ectopic vessel sprouting after radial glia ablation was dramatically reduced in the heat shocked *TgBAC(gfap:gal4ff);Tg(UAS:mcherry-NTR);Tg(kdrl:EGFP);Tg(hsp70l: sflt4, cryaa:cerulean)* larvae (which exhibit reduced numbers of vISVs). (G–J) 130 hpf *TgBAC(gfap:gal4ff);Tg(UAS:mcherry-NTR);Tg(kdrl:EGFP)* trunk vasculature visualized by *Tg(kdrl:*EGFP) expression. Animals were pre-treated with 2 mM Mtz between 30 and 54 hpf and then treated with DMSO (G), 2 µM Sunitinib (H), 2.5 µM SKLB1002 (I), or 10 µM LY294002 (J) between 54 and 130 hpf. As compared to DMSO, these chemicals significantly inhibited ectopic vessel sprouting after radial glia ablation. Scale bar, 100 µm. (K) Quantification of average number of somites that showed ectopic blood vessels in 130 hpf larvae that were treated with different chemicals after radial glia ablation (10 somites examined per animal). In all panels, values represent means ± SEM (*** indicates $p < 0.001$ by one-way ANOVA followed by Tukey's HSD test).

The following source data and figure supplement are available for figure 4:

**Source data 1.** Quantification of aISVs and vISVs number.

**Figure supplement 1.** Genetic inhibition of Vegfr2, but not Vegfr3 or Bmp, signaling blocks ectopic vessel sprouting after radial glia ablation.

## *sflt1* mutants exhibit ectopic venous ISV sprouting around the spinal cord

An endogenous protein that is known to act as a negative regulator of Vegf/Vegfr2 signaling is sFlt1. Previous studies have shown that sFlt1 functions as a Vegf decoy receptor and thereby limits angiogenic sprouting induced by Vegf/Vegfr2 signaling (Ruiz *de Almodovar et al., 2009*). We thus hypothesized that radial glia negatively modulate Vegf/Vegfr2 signaling via sFlt1 to prevent vISV over-sprouting. To test this hypothesis, we generated zebrafish *sflt1* mutants by targeted genome editing using the CRISPR/Cas9 system (*Figure 5A,B*, and *Figure 5—figure supplement 1*). We identified a mutation, *bns29*, which is predicted to lead to a premature stop codon at tyrosine residue 86 of both sFlt1 and a membrane-bound form of Flt1/Vegfr1 (mFlt1) (*Figures 5A* and *Figure 5—figure supplement 1*). We also used *flt1*$^{fh390}$ mutants, which are predicted to lack the transmembrane and tyrosine kinase domains of mFlt1 without affecting sFlt1 (*Rossi et al., 2016*), in order to determine the role of mFlt1. The genotypes of *flt1*$^{bns29}$ and *flt1*$^{fh390}$ animals were determined by high resolution melt analysis (*Figure 5B*). Interestingly, we observed that the *flt1*$^{fh390}$ mutants, which lack mFlt1 signaling function, do not exhibit any obvious vascular phenotypes in the trunk (*Figure 5I*; n = 20). However, we found that the *flt1*$^{bns29}$ mutants, which lack sFlt1 function, exhibit ectopic vessel sprouting specifically around the spinal cord (*Figure 5C–J*; n = 20–24 for each genotype), a phenotype similar to that found after radial glia ablation. Thus, phenotypic analyses of these different *flt1* mutants suggest that sFlt1, but not mFlt1, functions to prevent excessive sprouting of ISVs in the dorsal trunk. In addition, we found that ectopic vessel sprouting in *flt1*$^{bns29}$ mutants begins around 74 hpf, and that these ectopic vessels mostly derive from vISVs, as observed after radial glia ablation. As assessed by *Tg(-0.8flt1:tdTomato)* expression, ectopic sprouts in *flt1*$^{bns29}$ mutants mostly derive from vISVs at 100 hpf (*Figure 5K–K″*) and at 154 hpf exhibit weak *Tg(-0.8flt1:tdTomato)* expression as vISVs do (*Figure 5L–L″*), which is again very similar to what was observed after radial glia ablation.

## sFlt1 function in endothelial cells can limit venous over-sprouting around the spinal cord

To gain mechanistic insight into how radial glia and sFlt1 may cooperate to regulate this vascular patterning, we first characterized sFlt1 expression in detail. Previous work has shown *sflt1* expression primarily in ECs during early embryonic stages by *in situ* hybridization (*Krueger et al., 2011*; *Zygmunt et al., 2011*), and has also reported *sflt1* and/or *mflt1* expression in some neurons within the spinal cord by analyzing a BAC transgenic reporter line, *TgBAC(flt1:YFP)*$^{hu4624}$, in which YFP is inserted in a *flt1* BAC (*Hogan et al., 2009a*). We first carefully examined *TgBAC(flt1:YFP)* expression at stages when ectopic sprouting emerges in *flt1*$^{bns29}$ mutants, and observed that *TgBAC(flt1:YFP)* expression is indeed found in the vasculature as well as in some spinal cord cells (*Figure 6A and A′*). As suggested previously (*Krueger et al., 2011*), we observed that this *TgBAC(flt1:YFP)* expression within the spinal cord is in neurons, since genetic ablation of neurons resulted in an almost complete loss of this expression (*Figure 6A–B′*). In addition, we found that *TgBAC(flt1:YFP)* expression within the spinal cord did not co-localize with the radial glia mCherry reporter expression at 82 hpf (*Figure 6C–D″*; none of 51 *TgBAC(flt1:YFP)*-positive cells analyzed in 6 fish were positive for the radial glia mCherry reporter), suggesting that radial glia are unlikely to be a source of sFlt1 or mFlt1.

In order to address the cell-autonomy of sFlt1 function in patterning the veins around the spinal cord, we transplanted *flt1*$^{bns29}$ mutant cells with a GFP endothelial label, *TgBAC(etv2:EGFP)*$^{ci1}$, or *flt1*$^{+/+}$ cells with a BFP endothelial label, *Tg(kdrl:tagBFP)*$^{mu293}$, into WT or *sflt1* mutant animals, respectively (*Figure 6E*). We found that when *flt1*$^{bns29}$ mutant cells were transplanted into *flt1*$^{+/+}$ hosts, some transplanted mutant ECs, but not *flt1*$^{+/+}$ ECs, exhibited ectopic sprouting (*Figure 6F, G, and J*; 79 out of 179 transplanted dorsal ISVs exhibited ectopic sprouting in 31 fish analyzed). Conversely, when *flt1*$^{+/+}$ cells were transplanted into *flt1*$^{bns29}$ mutant hosts, most transplanted *flt1*$^{+/+}$ ECs did not show ectopic sprouting, in contrast to *flt1*$^{bns29}$ mutant ECs that exhibited clear sprouting (*Figure 6H,I, and J*; 137 out of 144 transplanted dorsal ISVs exhibited no ectopic sprouting in 33 fish analyzed). Thus, these transplantation data indicate that sFlt1 acts cell-autonomously in endothelial cells to limit ectopic sprouting.

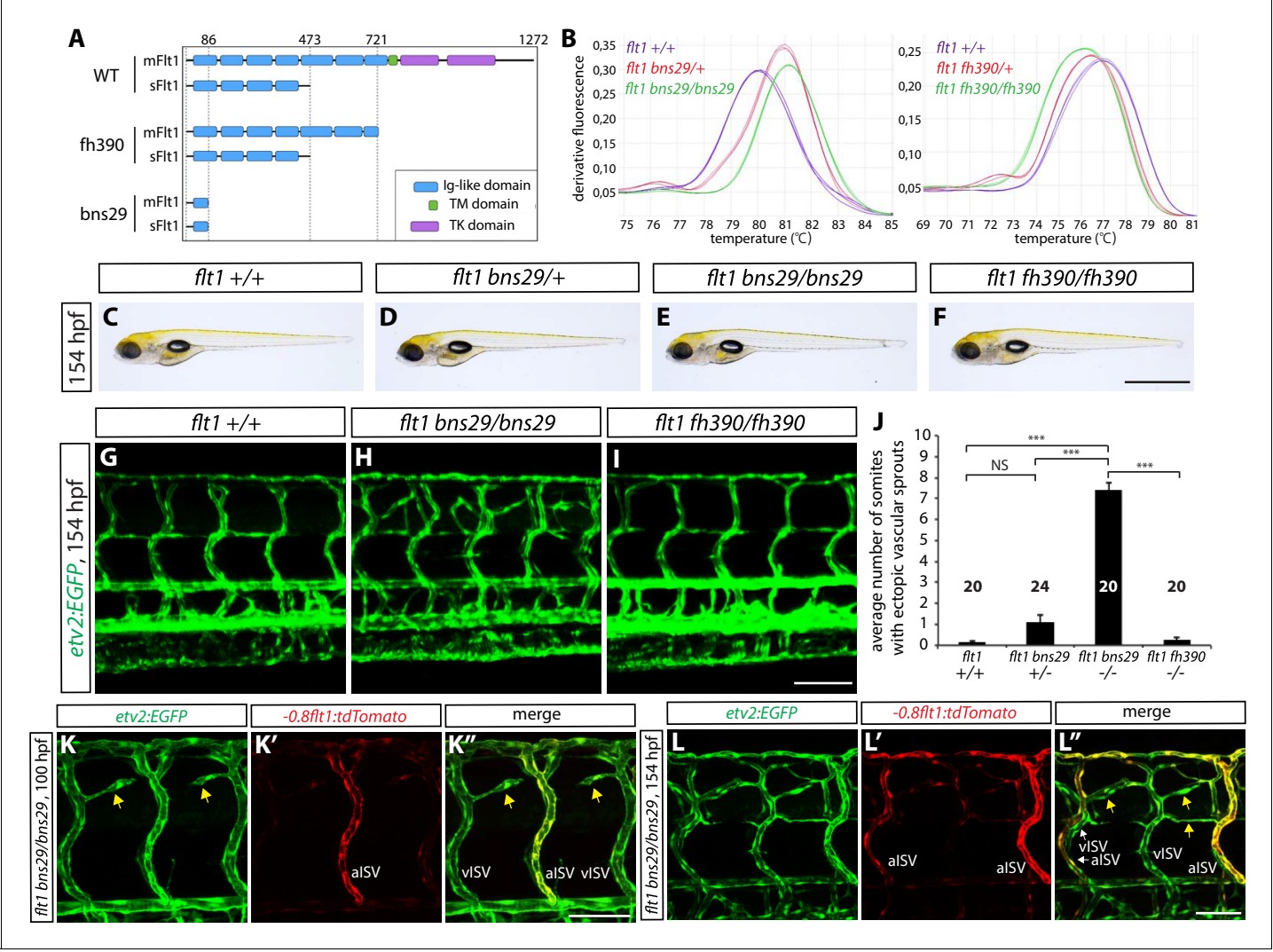

Figure 5. *sflt1* mutants exhibit ectopic venous ISV sprouting around the spinal cord. (A) Predicted domain structures of WT and mutant Flt1 (mFlt1 and sFlt1). (B) High resolution melt analysis used to determine the *flt1^bns29* (left panel) and *flt1^fh390* (right panel) genotypes. Melting curves for 3 independent fish of each genotype are shown. (C–F) Representative brightfield images of 154 hpf *flt1^+/+* (C), *flt1^bns29/+* (D), *flt1^bns29/bns29* (E), and *flt1^fh390/fh390* (F) larvae. Scale bar, 1 mm. (G–I) 154 hpf *TgBAC(etv2:EGFP) flt1^+/+* (G), *flt1^bns29/bns29* (H), and *flt1^fh390/fh390* (I) trunk vasculature. Ectopic vessel sprouting is observed in the dorsal trunk of *flt1^bns29/bns29* (H), but not *flt1^+/+* (G) or *flt1^fh390/fh390* (I), fish. Scale bar, 100 μm. (J) Quantification of average number of somites that showed ectopic blood vessels in *flt1^+/+*, *flt1^bns29/+*, *flt1^bns29/bns29*, and *flt1^fh390/fh390* larvae (10 somites examined per animal; ≥20 animals examined per genotype). (K–K") 100 hpf *TgBAC(etv2:EGFP);Tg(-0.8flt1:tdTomato) flt1^bns29/bns29* trunk vasculature visualized by *TgBAC(etv2*:EGFP) (K) and *Tg(-0.8flt1*:tdTomato) (K') expression. Ectopic vessel sprouts in *flt1^bns29/bns29* fish derive from venous ISVs in most cases (yellow arrows). Scale bar, 50 μm. (L–L") High magnification images of 154 hpf *TgBAC(etv2:EGFP);Tg(-0.8flt1:tdTomato) flt1^bns29/bns29* trunk vasculature. Ectopic vessels in the dorsal trunk exhibit weak *Tg(-0.8flt1*:tdTomato) expression as vISVs do (yellow arrows, L"). Scale bar, 50 μm. In all panels, values represent means ± SEM (*** indicates $p < 0.001$ by one-way ANOVA followed by Tukey's HSD test).

The following figure supplement is available for figure 5:

**Figure supplement 1.** Generation of zebrafish *flt1^bns29* mutant allele.

## Radial glia regulate *sflt1* expression

Based on the phenotypic similarities observed after radial glia ablation and in *flt1^bns29* mutants, we hypothesized that radial glia positively regulate sFlt1 function *in vivo*. We started to test this hypothesis by examining gene expression and performed qPCR analysis for *sflt1* and the other Vegf receptor genes on cDNA obtained from 100 hpf *Tg(gfap:NTR)* fish trunk mRNA after treatment with

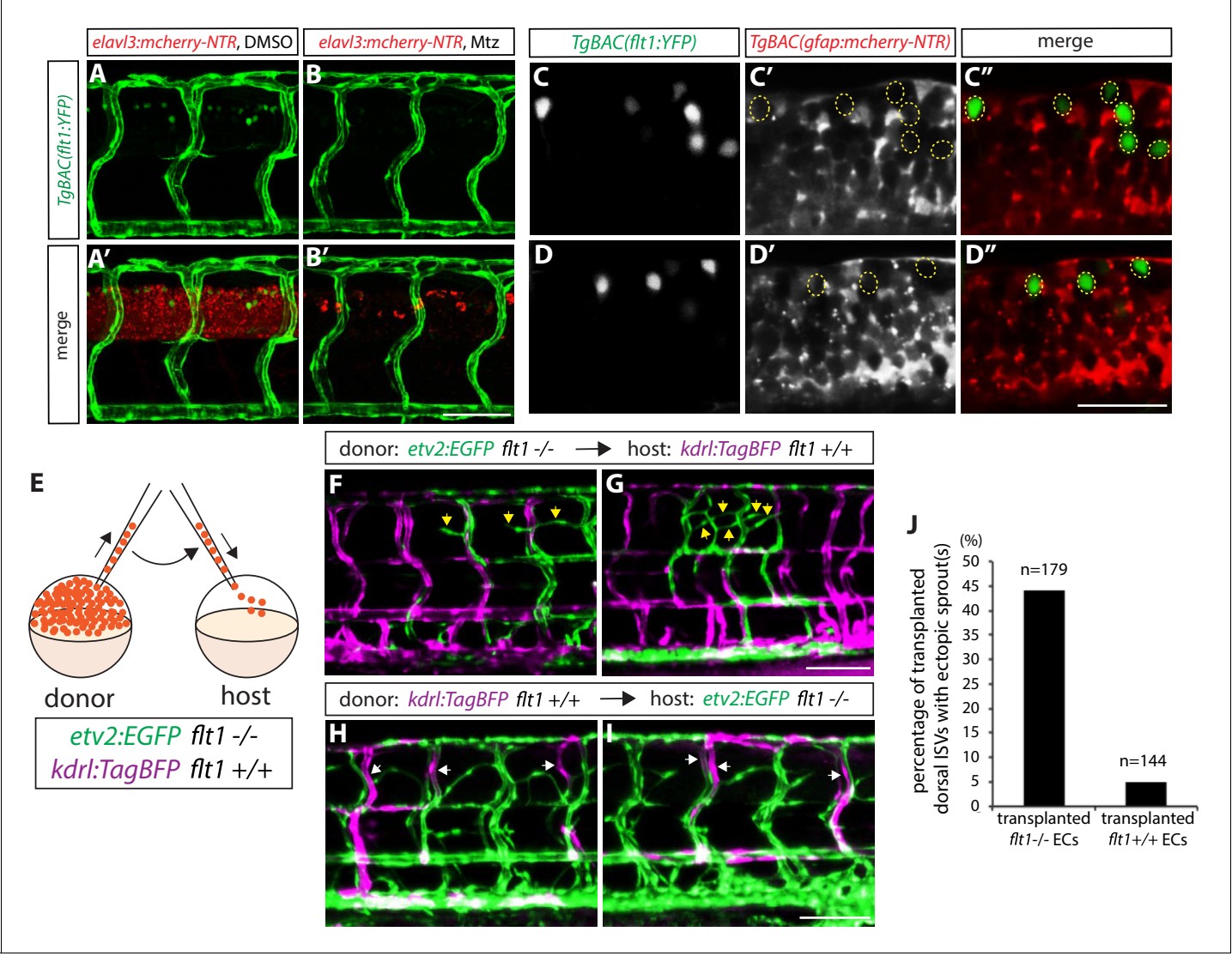

**Figure 6.** sFlt1 function in endothelial cells can limit venous ISV over-sprouting around the spinal cord. (A–B') 82 hpf *TgBAC(flt1:YFP);Tg(elavl3:gal4);Tg (UAS:mcherry-NTR)* trunks after treatment with DMSO (**A** and **A'**) or 2 mM Mtz (**B** and **B'**) between 30 and 82 hpf. *TgBAC(flt1:*YFP) expression is observed strongly in ECs and weakly in cells residing in the spinal cord. Genetic ablation of neurons leads to an almost complete loss of *TgBAC(flt1:* YFP) expression in the spinal cord (**B** and **B'**). Scale bar, 50 μm. (**C–C"** and **D–D"**) High magnification single-plane confocal images of 80 hpf *TgBAC(flt1:* YFP);TgBAC(gfap:gal4ff);Tg(UAS:mcherry-NTR)* spinal cord. *TgBAC(flt1:*YFP) expression does not co-localize with mCherry⁺ radial glia. Yellow dashed circles mark the position of the *TgBAC(flt1:*YFP)-positive spinal cord cells (**C', C", D'**, and **D"**). Scale bar, 30 μm. (**E**) Schematic diagram of the transplantation assay (**F–J**). Cells taken from *TgBAC(etv2:EGFP) flt1^{bns29/bns29}* or *Tg(kdrl:TagBFP) flt1^{+/+}* donor embryos were transplanted into *Tg(kdrl: TagBFP) flt1^{+/+}* or *TgBAC(etv2:EGFP) flt1^{bns29/bns29}* host embryos, respectively. (**F** and **G**) 154 hpf *Tg(kdrl:TagBFP) flt1^{+/+}* trunk vasculature with transplanted cells from *TgBAC(etv2:EGFP) flt1^{bns29/bns29}* embryos. Yellow arrows point to ectopic sprouts from transplanted *TgBAC(etv2:EGFP) flt1^{bns29/bns29}* ECs in the dorsal trunk. Only the transplanted *TgBAC(etv2:EGFP) flt1^{bns29/bns29}* ECs, but not the neighboring *Tg(kdrl:TagBFP) flt1^{+/+}* ECs, exhibit ectopic ISV sprouting. Scale bar, 100 μm. (**H** and **I**) 154 hpf *TgBAC(etv2:EGFP) flt1^{bns29/bns29}* trunk vasculature with transplanted cells from *Tg(kdrl: TagBFP) flt1^{+/+}* embryos. White arrows point to transplanted *Tg(kdrl:TagBFP) flt1^{+/+}* ECs in the dorsal trunk, which do not exhibit ectopic sprouting, unlike neighboring *TgBAC(etv2:EGFP) flt1^{bns29/bns29}* ECs that show ectopic sprouting. Scale bar, 100 μm. (**J**) Percentage of transplanted ISVs in the dorsal trunk that exhibited ectopic sprout(s) at 154 hpf for the experiments shown in panels (**F–I**).

DMSO or Mtz starting at 30 hpf. We found that *sflt1* and *mflt1*, but not *vegfr2* or *vegfr3*, expression was significantly reduced in the Mtz-treated animals (*Figure 7A*). In contrast, when we compared gene expression in 148 hpf *Tg(gfap:NTR)* trunk cDNA after treatment with DMSO or Mtz starting at 78 hpf, we did not find significant differences in *sflt1* and *mflt1* expression, as compared to controls

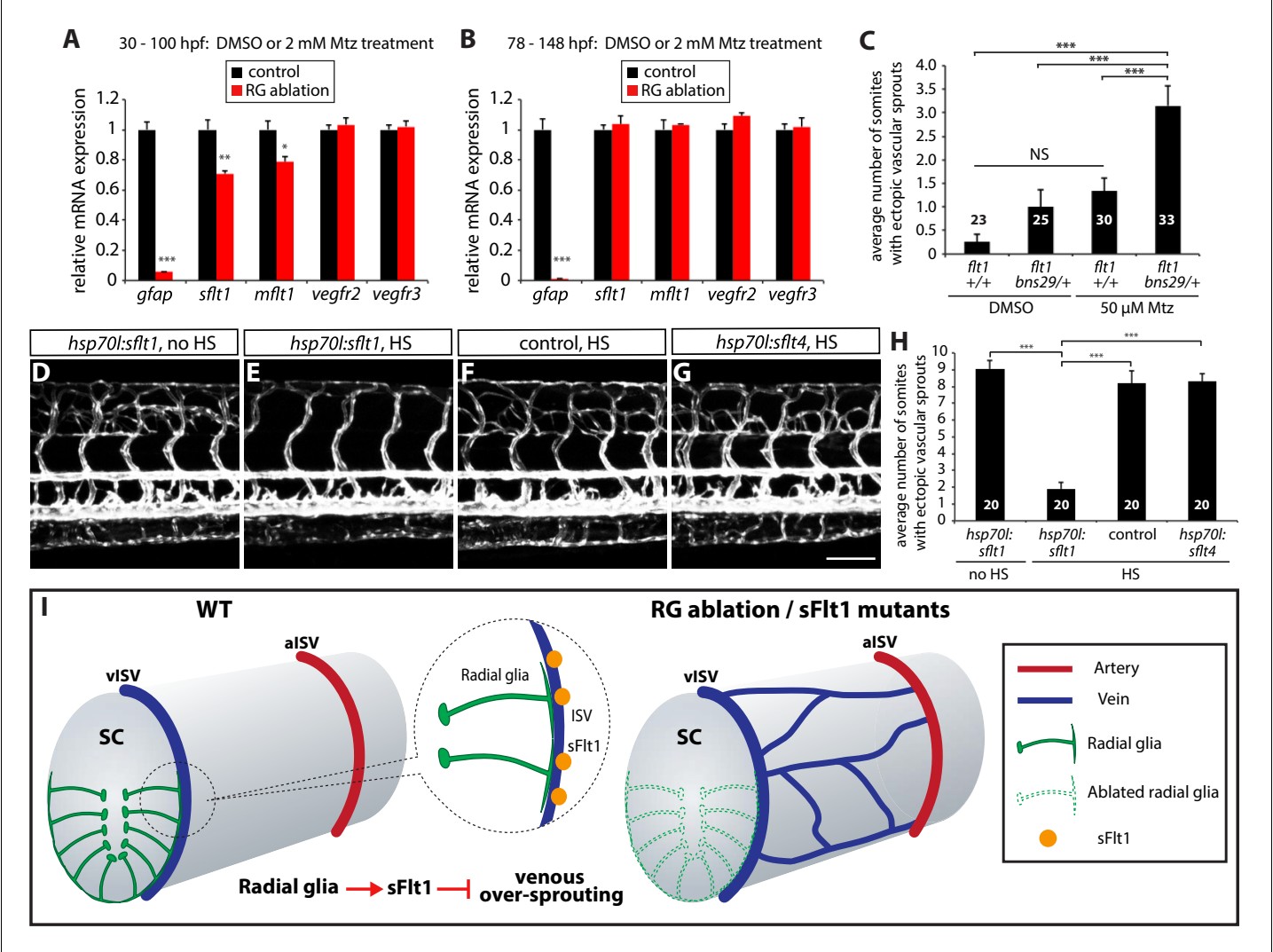

**Figure 7.** Radial glia limit venous over-sprouting around the spinal cord via the regulation of *sflt1* expression. (**A**) qPCR analyses of 100 hpf *TgBAC (gfap:gal4ff);Tg(UAS:mcherry-NTR)* trunk samples after treatment with DMSO or 2 mM Mtz between 30 and 100 hpf. Radial glia-ablated fish trunk samples show significantly reduced mRNA expression of *gfap, sflt1,* and *mflt1*. (**B**) qPCR analyses of 148 hpf *TgBAC(gfap:gal4ff);Tg(UAS:mcherry-NTR)* trunk samples after treatment with DMSO or 2 mM Mtz between 78 and 148 hpf. Radial glia-ablated fish trunk samples show significantly reduced *gfap* mRNA expression; however, *sflt1* and *mflt1* mRNA expression was not significantly different between control and radial glia-ablated samples. (**C**) Quantification of average number of somites that showed ectopic sprouting in 154 hpf *flt1*$^{+/+}$ and *flt1*$^{bns29/+}$ larvae treated with DMSO or 50 μM Mtz between 30 and 154 hpf (10 somites examined per animal; ≥23 animals examined per condition). *flt1*$^{+/+}$ larvae treated with 50 μM Mtz or *flt1*$^{bns29/+}$ larvae treated with DMSO did not show a statistically significant change in ectopic ISV sprouting compared to *flt1*$^{+/+}$ larvae treated with DMSO. However, *flt1*$^{bns29/+}$ larvae treated with 50 μM Mtz exhibited a significantly increased number of somites with ectopic sprouts compared to the other three treatments. See also *Figure 7—source data 1* for quantification. (**D–G**) 154 hpf *TgBAC(gfap:gal4ff);Tg(UAS:mcherry-NTR);Tg(kdrl:EGFP);Tg (hsp70l:sflt1, cryaa:cerulean)* (**D** and **E**), *TgBAC(gfap:gal4ff);Tg(UAS:mcherry-NTR);Tg(kdrl:EGFP)* (**F**), and *TgBAC (gfap:gal4ff);Tg(UAS:mcherry-NTR);Tg(kdrl:EGFP);Tg(hsp70l:sflt4, cryaa:cerulean)* (**G**) trunk vasculature visualized by *Tg(kdrl:*EGFP) expression. Animals were treated with 2 mM Mtz between 30 and 54 hpf and then subject to no heat shock (**D**) or multiple heat shocks (**E–G**) starting at 62 hpf and every 12 hr after that. Overexpression of sFlt-1 blocks ectopic vessel sprouting after radial glia ablation (**E**). Scale bar, 100 μm. (**H**) Quantification of average number of somites that showed ectopic sprouting for the experiments shown in panels **D–G** (10 somites examined per animal; 20 animals examined per condition). Quantification was performed at 154 hpf. Overexpression of sFlt-1, but not of sFlt-4, significantly inhibited ectopic vessel sprouting after radial glia ablation. HS: heat shock. (**I**) Schematic diagrams showing radial glia regulation of the vascular patterning around the spinal cord. During development, the end-feet of radial glia lie in close proximity to the dorsal ISVs which surround the spinal cord. Genetic ablation of radial glia in early embryos leads to selective over-sprouting of venous ISVs. Perturbation of sFlt1 function also leads to over-sprouting of vISVs, and radial glia regulate the precise patterning of the venous vasculature around the spinal cord at least in part via the control of sFlt1 function in endothelium. RG: radial glia. In panels (**C** and **H**), values represent means ± SEM (*** indicates p<0.001 by one-way ANOVA followed by Tukey's HSD test). In panels (**A** and **B**), values represent means ± SEM (*, **, and *** indicate p<0.05, p<0.01, and p<0.001, respectively, by Student's *t* test).

*Figure 7 continued on next page*

*Figure 7 continued*

The following source data is available for figure 7:

**Source data 1.** Quantification of average number of somites that showed ectopic blood vessels in *flt1⁺/⁺* and *flt1ᵇⁿˢ²⁹/⁺* larvae.

(*Figure 7B*). These data are consistent with our observations that radial glia ablation in early embryos, but not in later larval stages (data not shown), leads to excessive sprouting of vISVs, and indicate that radial glia regulate *sflt1* expression during embryonic stages. Given the previous reports that *sflt1* is primarily expressed in ECs during embryonic stages (*Krueger et al., 2011*; *Zygmunt et al., 2011*), it is likely that radial glia regulate *sflt1* expression in a cell non-autonomous manner.

We next asked whether the s*flt1* mutation affected the sensitivity of vISVs after radial glia ablation by comparing vessel sprouting in *flt1⁺/⁺* and *flt1ᵇⁿˢ²⁹/⁺* fish. We used a lower concentration of Mtz (50 μM) for these experiments to reduce the extent of ISV over-sprouting after radial glia ablation (*Figure 7C*; n = 23–30). Notably, we found that *flt1ᵇⁿˢ²⁹/⁺* fish exhibit a markedly increased degree of ISV over-sprouting after partial radial glia ablation as compared to controls (*Figure 7C* and *Figure 7—source data 1*; n = 33), providing evidence that radial glia ablation and sFlt1 functionally interact. We also found that overexpression of sFlt1, but not sFlt4, fully blocked the ectopic ISV sprouting after radial glia ablation, thereby normalizing vascular patterning around the spinal cord (*Figure 7D–H*; n = 20 for each group). These findings show that sFlt1 overexpression is sufficient to rescue the ectopic vISV sprouting phenotype after radial glia ablation, and suggest that sFlt1 acts downstream of radial glia in the regulation of this vascular patterning process.

## Discussion

In this work, we identified previously unrecognized, yet critical, roles for CNS-resident progenitors in orchestrating the vascular architecture around the CNS. Our findings reveal that radial glia within the CNS act as negative as well as selective angiogenic regulators that determine the precise patterning of the venous vasculature around the developing spinal cord (*Figure 7I*). Specifically, we found that radial glia ablation in zebrafish embryos leads to the over-sprouting of venous ISVs around the spinal cord, a process driven by Vegf/Vegfr2 signaling; we also found that *sflt1* mutants recapitulate the vascular phenotype observed after radial glia ablation, and that sFlt1 acts downstream of radial glia and functions in endothelial cells to limit venous over-sprouting around the spinal cord. Altogether, these findings suggest a model whereby radial glia positively regulate sFlt1 function in endothelial cells to establish the precise venous patterning around the spinal cord during embryonic and early larval development, providing critical and novel insights into the cellular and molecular mechanisms by which the vascular networks around developing organs are patterned.

Many questions emerge from these findings. First, we speculate that the regulation of vascular patterning around the developing CNS by radial glia may constitute a means to ensure optimal vascularization for CNS function, and thus, determining the functional relevance of the observed vascular defects is one of the critical next steps. These findings indicate more generally that progenitor cells in developing organs may function as critical regulators to establish the microenvironment for the mature organs derived from these progenitors. Second, it will be of particular interest to the neuroscience and vascular biology fields to further investigate how CNS radial glia differentially modulate arterial and venous ECs. Recent studies have discovered a lymphatic network lining the meningeal dura mater of the mouse brain (*Aspelund et al., 2015*; *Louveau et al., 2015*), and whose architecture is very distinct from that of the arterial and venous networks in the CNS. Our identification of spinal cord radial glia as negative as well as selective angiogenic regulators that specifically affect venous vessels is intriguing in light of previous studies that revealed neuronal progenitors as positive angiogenic regulators that initiate arterial sprouting into neural tissues (*Kurz et al., 1996*; *Haigh et al., 2003*; *Kurz, 2009*). These findings may indicate that CNS-resident progenitors are able to differentially regulate the patterning of these distinct vessel networks within as well as around the CNS.

How do CNS radial glia and sFlt1 selectively regulate venous ECs around the developing spinal cord? One hypothesis is that the distinct density and/or characteristics of perivascular cells that cover arteries and veins play a role in this process. Indeed, it was recently shown that in zebrafish, mural cells emerge around the ISVs starting at 48 hpf and that they preferentially cover aISVs (*Ando et al., 2016*). Preferential coverage of arteries by mural cells was also observed in mouse embryos (*Hellstrom et al., 1999*). This differential coverage of arterial and venous ISVs by mural cells may explain the near exclusive over-sprouting of vISVs after radial glia ablation and in *sflt1* mutants. Perhaps, coverage of aISVs by mural cells blocks most communication between radial glia and arterial ECs. To begin to test this hypothesis, we carefully analyzed *pdgfrb^um148* mutants (*Kok et al., 2015*), which should exhibit a slightly decreased number of ISV-covering mural cells as previously shown in *pdgfrb^sa16389* mutants (*Ando et al., 2016*), after ablating radial glia starting at 30 hpf. However, we did not observe a significantly increased number of over-sprouting aISV in these experiments (data not shown). We speculate that apparently decreased, yet incompletely eliminated, ISV-covering mural cells in these *pdgfrb* mutant animals (*Kok et al., 2015*; *Ando et al., 2016*) could still protect aISVs from over-sprouting after radial glia ablation. Alternative approaches that allow complete ablation of ISV-covering mural cells will be needed to address this issue in a more conclusive manner.

How do radial glia regulate *sflt1* expression? Previous work indicates that Sema/PlxnD1 signaling positively regulates *sflt1* expression in zebrafish embryos (*Zygmunt et al., 2011*). It was shown that *sflt1* transcripts detected with a *sflt1*-specific riboprobe were markedly decreased in *plxnd1^fov01b* mutants and that *sflt1* transcript levels measured by qPCR were also reduced in *plxnd1^fov01b/+* embryos compared to WT (*Zygmunt et al., 2011*). Although we did not observe ectopic sprouting from the DA in *flt1^bns29* mutants as was seen in *plxnd1^fov01b* mutants (*Zygmunt et al., 2011*), it is still possible that this signaling pathway mediates the negative angiogenic effects of radial glia and regulates *sflt1* expression to limit venous over-sprouting around the spinal cord. Continuous overexpression of dominant-negative PlxnD1 in ECs from late embryonic stages (around 2.5 dpf) may allow us to test this hypothesis. Regarding sFlt1 function, our genetic mosaic analyses show that sFlt1 function in ECs can limit venous over-sprouting around the spinal cord. However, *TgBAC(flt1:YFP)* expression is also observed in small subsets of spinal cord neurons (*Figure 6A–B'*)(*Krueger et al., 2011*). Although we did not observe vISV over-sprouting in neuron-ablated fish (*Figure 2D–F*), which show a loss of *TgBAC(flt1:YFP)* expression in the spinal cord (*Figure 6A–B'*), it is possible that Flt1 in spinal cord neurons may also contribute to the vISV over-sprouting phenotype observed in *flt1^bns29* mutants. Finally, whether the communication between radial glia and venous ECs involves secreted factors and/or direct cell-cell communication remains a major question to resolve.

In summary, our observations of vascular pattern regulation within mesodermal tissues by adjacent ectoderm-derived CNS radial glia suggest a mechanism by which tissue vascularization is coordinated by adjacent tissue interactions. The interaction between factors generated by the ectoderm and factors deriving from resident cells in the mesoderm opens a new avenue for future studies on the coordination of tissue vascularization; and it will be of broad interest to determine whether and how vascularization events between different tissues (e.g., endoderm-derived organs and mesodermal tissues) are coordinated during development. Therefore, our identification of a critical role for CNS-resident progenitors in patterning the vasculature around the CNS will advance our understanding of the mechanisms governing vascular network formation, in particular how distinct vessel networks, as well as vascularization patterns, around developing organs are formed.

## Materials and methods

### Zebrafish husbandry and transgenic lines

All zebrafish husbandry was performed under standard conditions in accordance with institutional (UCSF and MPG) and national ethical and animal welfare guidelines. *Tg(kdrl:EGFP)^s843* (*Jin et al., 2005*), *Tg(kdrl:NLS-EGFP)^ubs1* (*Blum et al., 2008*), *Tg(kdrl:Has.HRAS-mcherry)^s896* (*Chi et al., 2008*), *Tg(fli1a:EGFP)^y1* (*Lawson and Weinstein, 2002*), *TgBAC(etv2:EGFP)^ci1* (*Proulx et al., 2010*), *Tg(-0.8flt1:tdTomato)^hu5333* (*Bussmann et al., 2010*), *TgBAC(flt1:YFP)^hu4624* (*Hogan et al., 2009a*), *Tg(elavl3:gal4-vp16)^psi1* (*Stevenson et al., 2012*), *TgBAC(gfap:gfap-gfp)^zf167* (*Lam et al., 2009*), *Tg(gfap:GFP)^mi2001* (*Bernardos and Raymond, 2006*), *Tg(elavl3:EGFP)^knu3* (*Park et al., 2000*), *Tg(UAS-E1b:NfsB-mCherry)^c264* (*Davison et al., 2007*), *Tg(mpx:gal4)^sh267* (*Robertson et al., 2014*), *Tg(-*

*4.9sox10:EGFP)^{ba2}* (*Carney et al., 2006*), *Tg(hsp70l:nog3)^{fr14}* (*Chocron et al., 2007*), *irf8^{st96}* (*Shiau et al., 2015*), *pdgfrb^{um148}* (*Kok et al., 2015*), *erbb2^{st61}* (*Lyons et al., 2005*), and *flt1^{fh390}* (*Rossi et al., 2016*) were used in this study. Fish embryos/larvae were raised at 28°C.

## Generation of new transgenic lines

A *TgBAC(gfap:gal4ff)^{s995}* fish line was generated by following the standard BAC recombineering protocol described previously (*Bussmann and Schulte-Merker, 2011*). Briefly, the CH211-154O10 BAC clone (BACPAC Resource Center) was engineered to replace the *gfap* start codon with *gal4ff*. The engineered construct was injected into one-cell stage embryos together with transposase mRNA transcribed *in vitro*. The *Tg(hsp70l:sflt1, cryaa-cerulean)^{bns80}* and *Tg(hsp70l:sflt4, cryaa-cerulean)^{bns82}* fish lines were generated by inserting zebrafish *sflt1* and human *sFLT4-IgG-Fc* (*Makinen et al., 2001*), respectively, under the zebrafish *hsp70l* promoter (*Halloran et al., 2000*). The eye-marker cassette, *cryaa:cerulean* (*Kurita et al., 2003*; *Hesselson et al., 2009*), was also introduced in the same construct. These constructs were then injected into one-cell stage embryos together with I-Sce I meganuclease (NEB) supplemented with 1X cutsmart NEB buffer. The *Tg(kdrl:tagBFP)^{mu293}* fish line was generated by cloning the sequence encoding tagBFP downstream of a 7.6 kb *kdrl* promoter flanked by tol2 sites. *kdrl:tagBFP* plasmid DNA was injected together with transposase mRNA.

## Generation and genotyping of *flt1* mutants

The *flt1^{bns29}* mutant allele was generated by targeted genome editing using the CRISPR/Cas9 system as follows: We targeted the third exon of *flt1*, which encodes part of the 1st Ig-like domain. The *flt1^{bns29}* mutant allele harbors a 1 base pair deletion and a 3 base pair insertion. The *flt1^{bns29}* allele (c.258delinsGGC) is predicted to lead to a premature stop codon at tyrosine residue 86 and thus encode a truncated polypeptide containing a stretch of 85 amino acids of both sFlt1 and mFlt1. The Cas9 vector (pT3TS-nlsCas9nls) was purchased from Addgene. Cas9 mRNA was prepared *in vitro* using the mMESSAGE mMACHINE T3 transcription kit (Ambion) after linearizing the plasmid with XbaI (NEB). The mRNA was purified using RNA Clean & Concentrator-5 kit (Zymo Research). The gRNA was designed using Optimized CRISPR Design algorithm (http://crispr.mit.edu). The oligos (taggGTGTCAGCCGGCTGCAATAC and aaacGTATTGCAGCCGGCTGACAC) were annealed and ligated into the gRNA plasmid (pT7-gRNA vector) after digesting the plasmid with BsmbI (NEB). The gRNA was prepared *in vitro* using the MEGAshortscript T7 transcription kit (Ambion) after linearizing the plasmid with BamHI-HF (NEB). The gRNA was purified using RNA Clean & Concentrator-5 kit. 1 nL of a solution containing 250 ng/µl of Cas9 mRNA and 100 ng/µl of gRNA was injected at the one-cell stage.

WT, heterozygous, and homozygous *flt1^{bns29}* or *flt1^{fh390}* animals were identified by high-resolution melt analysis of PCR using the following primers:

```
flt1 bns29 Fw: 5'- ACTCAGTGCGGGAAGAAAAG -3'
flt1 bns29 Rv: 5'- CCCGTGTGCTGAGTTAAAGC -3'
flt1 fh390 Fw: 5'- CAGTTGAAGACCAGGGATTTT -3'
flt1 fh390 Rv: 5'- CCACAGACCCCCTCTGATT -3'
```

## High-resolution melt analysis

An Eco Real-Time PCR System (Illumina) was used for the PCR reactions and high-resolution melt analysis. DyNAmo SYBR green (Thermo Fisher Scientific, Germany) was used in these experiments. PCR reaction protocols were 95°C for 15 s, then 40 cycles of 95°C for 2 s, 60°C for 2 s, and 72°C for 2 s. Following the PCR, a high-resolution melt curve was generated by collecting SYBR-green fluorescence data in the 65–95°C range. The analyses were performed on normalized derivative plots.

## Metronidazole (Mtz) treatment

Mtz substrate (Sigma, M3761) was used at 2 mM dissolved in egg water containing 1% DMSO for most of the cell ablation experiments in this study. Prior to treatment with 2 mM Mtz or 1% DMSO control, embryos were manually dechorionated with forceps and then incubated with freshly

prepared 2 mM Mtz or 1% DMSO in egg water. To wash away the Mtz, fish embryos/larvae were washed twice in dishes containing egg water. Although a minority of the animals developed prominent pericardial edema and/or severe gross anatomical abnormalities after radial glia or neuronal ablation, most did not exhibit such abnormalities except non-inflated swim bladders and bent tails. We analyzed only the animals that did not exhibit pericardial edema or severe gross anatomical abnormalities. Treatment of *flt1*[+/+] and *flt1*[bns29/+] fish with Mtz was performed as follows: animals were treated with 50 μM Mtz or 1% DMSO between 30 and 154 hpf after manual dechorionation.

## qPCR

qPCR was performed on cDNA obtained from *TgBAC(gfap:gal4ff)*[s995];*Tg(UAS-E1b:NfsB-mCherry)*[c264] larval trunk mRNA after treatment with 1% DMSO or 2 mM Mtz. Animals were treated with 1% DMSO or 2 mM Mtz between 30 and 100 hpf or between 78 and 148 hpf. An Eco Real-Time PCR System was used for qPCR experiments, and gene expression levels were normalized relative to that of a reference gene, *elfα*. All reactions were performed in technical duplicates, and the results represent 4 independent biological samples (15 larval trunks pooled for each sample) including the SEM. *Table 1* shows the primers used for these experiments.

## Confocal and stereo microscopy

An LSM 700 confocal laser scanning microscope (Zeiss) was used for live and immunofluorescence imaging. Fish embryos and larvae were anaesthetized with a low dose of tricaine, placed in a glass-bottom Petri dish (MatTek) with a layer of 1.2% low melt agarose, and imaged using W N-ACP 20X/0.5 and W Plan-Apochromat 40×/1.0 objective lenses. Immunostained fish sections were imaged using a C Apo 40X/1.1 objective lens. For FITC-Dextran microangiography, fluorescein isothiocyanate (FITC)-dextran, 2000 kDa (Sigma) was injected into the common cardinal vein and imaged after 10 min. An SMZ 25 stereomicroscope (Nikon) was used for brightfield images of anaesthetized fish.

## Heat shock treatments

Fish embryos/larvae raised at 28°C were subject to 37°C heat shock for 1 hr by replacing the egg water with pre-warmed (37°C) egg water and then kept in a 37°C incubator. After each heat shock, the fish embryos/larvae were kept at room temperature for 10 min to cool down and then transferred to a 28°C incubator. Heat shock experiments after radial glia ablation were performed as follows: *TgBAC(gfap:gal4ff)*[s995];*Tg(UAS-E1b:NfsB-mCherry)*[c264];*Tg(kdrl:EGFP)*[s843];*Tg(hsp70l:sflt1, cryaa-cerulean)*[bns80] and *TgBAC(gfap:gal4ff)*[s995];*Tg(UAS-E1b:NfsB-mCherry)*[c264];*Tg(kdrl:EGFP)*[s843];*Tg(hsp70l:sflt4, cryaa-cerulean)*[bns82] animals were treated with 2 mM Mtz between 30 and 54 hpf, washed twice in egg water, and then subject to one heat shock every 12 hr starting at 62 until 154 hpf. Heat shock experiments that reduce the number of vISVs were performed as follows: *TgBAC(gfap:gal4ff)*[s995];*Tg(UAS-E1b:NfsB-mCherry)*[c264];*Tg(kdrl:EGFP)*[s843];*Tg(hsp70l:sflt4, cryaa-cerulean)*[bns82] animals were given a heat shock at 29 hpf, then treated with 2 mM Mtz starting at 30 hpf.

**Table 1.** List of primers used for qPCR analysis.

| Primer name | Primer sequence (5' to 3') |
| --- | --- |
| *gfap* qPCR Fw | AGGCTCATGTGAAGAGGAGCATAG |
| *gfap* qPCR Rv | CCTCATTATGGCAGATCCTTCCTC |
| *sflt1* qPCR Fw (*sflt1* specific) | CGTCCCACCACCTCAAATCCAAT |
| *sflt1* qPCR Rv (*sflt1* specific) | AAGCCCATCCAGCCGCTATCAG |
| *mflt1* qPCR Fw (*mflt1* specific) | GCTCATTCAGGTGAAGTGGACAG |
| *mflt1* qPCR Rv (*mflt1* specific) | AGAAGATCGCCTTCATAATGTGG |
| *vegfr2* qPCR Fw | CATGAAGTGTCCACATATGTTTTTG |
| *vegfr2* qPCR Rv | TTTTGGTTTGGGTATGTTGTTCTAC |
| *vegfr3* qPCR Fw | GACCCAGAGCATCCATTCAT |
| *vegfr3* qPCR Rv | AGGCTCTGGATACGGCACTA |

Mtz-treated fish embryos were then heat shocked at 36 and 43 hpf for 1 hr by transferring the dishes containing a 2 mM Mtz solution from a 28°C incubator to a 37°C incubator. After the heat shock at 43 hpf, the embryos were treated with freshly prepared 2 mM Mtz until 100 hpf, washed twice in egg water, and then analyzed at 154 hpf.

## Pharmacological inhibitor treatments

Pharmacological experiments after radial glia ablation were performed as follows: *TgBAC(gfap: gal4ff)$^{s995}$;Tg(UAS-E1b:NfsB-mCherry)$^{c264}$; Tg(kdrl:EGFP)$^{s843}$* embryos were treated with 2 mM Mtz between 30 and 54 hpf, washed twice in dishes containing egg water, and then treated with pharmacological inhibitors from 54 until 130 hpf, replacing the drug solution every 24 hr. Final concentrations of the chemicals used were as follows: 2 μM sunitinib (Sigma), 2.5 μM SKLB1002 (Calbiochem), 10 μM LY294002 (Calbiochem), 5 μM DMHI (Calbiochem), 15 μM AG1295 (Calbiochem), and 3 μM PD158780 (Tocris). All listed chemicals were first prepared as 100X stock solutions dissolved in 100% DMSO and then diluted 100 times in egg water shortly before use.

## Cell transplantation

Cells were taken from *TgBAC(etv2:EGFP)$^{ci1}$ flt1$^{bns29/bns29}$* or *Tg(kdrl:tagBFP)$^{mu293}$ flt1$^{+/+}$* donor embryos at midblastula stages and transplanted into host embryos of the same age. Hosts were then imaged and analyzed at 154 hpf. For quantification, transplanted ECs that contributed to dorsal ISVs were analyzed for the presence of ectopic sprout(s) (each dorsal ISV that contained at least one transplanted EC was defined as a transplanted dorsal ISV). Note that when *TgBAC(etv2:EGFP) flt1$^{bns29/bns29}$* cells were transplanted into *Tg(kdrl:tagBFP) flt1$^{+/+}$* host embryos, we observed that approximately 5% of the transplanted hosts exhibited some mosaic EGFP expression in their spinal cord since *TgBAC(etv2:EGFP)$^{ci1}$* animals display non-specific EGFP expression in the neural tube of their trunk and tail (**Proulx et al., 2010**). We did not use animals with spinal cord EGFP expression in order to focus the analysis on ECs.

## Quantification of ectopic vessels

*Tg(kdrl:EGFP)$^{s843}$* or *TgBAC(etv2:EGFP)$^{ci1}$* EC reporter fish were used to quantify ectopic sprouts throughout this study. For the quantification of ectopic sprouts after radial glia or neuronal ablation, *Tg(kdrl:EGFP)$^{s843}$* fish were crossed to *TgBAC(gfap:gal4ff)$^{s995}$;Tg(UAS-E1b:NfsB-mCherry)$^{c264}$* or *Tg (elavl3:gal4-vp16)$^{psi1}$;Tg(UAS-E1b:NfsB-mCherry)$^{c264}$* fish, respectively. The 11 ISVs directly anterior to the anal opening were used for quantification (defined as 10 somites), and the number of somites that showed ectopic blood vessels in either side of the body was quantified and averaged.

## Quantification of endothelial cell numbers

*Tg(kdrl:NLS-EGFP)$^{ubs1}$* reporter fish were used to quantify EC number throughout this study. For the quantification of EC number after radial glia or neuronal ablation, *Tg(kdrl:NLS-EGFP)$^{ubs1}$* fish were crossed to *TgBAC(gfap:gal4ff)$^{s995}$;Tg(UAS-E1b:NfsB-mCherry)$^{c264}$* or *Tg(elavl3:gal4-vp16)$^{psi1}$;Tg (UAS-E1b:NfsB-mCherry)$^{c264}$* fish, respectively. The 5 ISVs directly anterior to the anal opening in both sides of the body were used to quantify the number of ECs in the dorsal and ventral parts of the trunk. Confocal *z*-stack images that span the ISVs in both sides of the body were taken, and cell numbers were manually determined using the ImageJ plugins.

## Quantification of ectopic sprouts that derive from aISVs or vISVs

Quantification of ectopic sprouts that derive from aISVs or vISVs after radial glia ablation was performed as follows: *TgBAC(gfap:gal4ff)$^{s995}$;Tg(UAS-E1b:NfsB-mCherry)$^{c264}$; Tg(kdrl:EGFP)$^{s843}$;Tg(- 0.8flt1:tdTomato)$^{hu5333}$* animals were treated with 2 mM Mtz between 30 and 100 hpf and then analyzed under the LSM 700 confocal microscope using a W Plan-Apochromat 40×/1.0 objective lens. Assignment of ectopic sprouts deriving from aISVs or vISVs was based on *Tg(-0.8flt1:*tdTomato) differential expression in aISVs and vISVs. As previously described (**Bussmann et al., 2010**), ISVs that were strongly labeled by *Tg(-0.8flt1:*tdTomato) expression were judged as aISVs, whereas those weakly labeled were judged as vISVs. In addition to using this differential *Tg(-0.8flt1:*tdTomato) expression between aISVs and vISVs, the identity of aISVs and vISVs was further confirmed by

examining their connections with axial vessels, namely the dorsal aorta and posterior cardinal vein, respectively. The 11 ISVs directly anterior to the anal opening were used for quantification.

## Analysis of *TgBAC(flt1:*YFP) expression

To analyze *TgBAC(flt1:*YFP)-positive spinal cord cells, *TgBAC(flt1:YFP)*$^{hu4624}$ reporter fish were crossed to *TgBAC(gfap:gal4ff)*$^{s995}$;*Tg(UAS-E1b:NfsB-mCherry)*$^{c264}$ fish. High magnification single-plane confocal images of 80 hpf *TgBAC(flt1:YFP)*$^{hu4624}$;*TgBAC(gfap:gal4ff)*$^{s995}$;*Tg(UAS-E1b:NfsB-mCherry)*$^{c264}$ spinal cords were taken using a W Plan-Apochromat 40×/1.0 objective lens and analyzed for the co-localization of YFP$^+$ spinal cord cells and mCherry$^+$ radial glia. To analyze *TgBAC(flt1:*YFP) expression after radial glia or neuronal ablation, *TgBAC(flt1:YFP)*$^{hu4624}$ fish were crossed to *TgBAC(gfap:gal4ff)*$^{s995}$;*Tg(UAS-E1b:NfsB-mCherry)*$^{c264}$ or *Tg(elavl3:gal4-vp16)*$^{psi1}$;*Tg(UAS-E1b:NfsB-mCherry)*$^{c264}$ fish, respectively. Animals were treated with 1% DMSO or 2 mM Mtz between 30 and 82 hpf, and imaged at 82 hpf. After radial glia ablation, no obvious differences in YFP expression were observed.

## Immunohistochemistry

Immunohistochemistry was performed as previously described (*Matsuoka et al., 2011*). The following primary antibodies were used: chicken anti-GFP (Aves Labs at 1:1000), rabbit anti-DsRed (Clontech at 1:300), and mouse anti-HuC/HuD (Invitrogen at 1:200).

## Statistical analysis

Statistical differences for mean values among multiple groups were determined using a one-way analysis of variance (ANOVA) followed by Tukey's multiple comparison test. Statistical differences for mean values between two groups were determined using Student's *t*-test. The criterion for statistical significance was set at $p < 0.05$. Error bars are SEM.

# Acknowledgements

We thank Herwig Baier, Cecilia Moens, William Talbot, and Stefan Schulte-Merker for kindly providing us with fish lines, Kari Alitalo for the human *sFLT4-IgG-Fc* plasmid, Ferdinand le Noble for discussions, Radhan Ramadass for help with the imaging, Petra Neeb and Rebecca Lee for technical assistance, Arica Beisaw, Hyun-Taek Kim and Hyouk-Bum Kwon for comments on the manuscript, and other members of the Stainier lab for sharing reagents and discussions. This work was supported by postdoctoral fellowships from the Damon Runyon Cancer Research Foundation (DRG#2104-12), Human Frontier Science Program (LT001023/2012-L), and Japan Society for the Promotion of Science to RLM, postdoctoral start-up grants from the Excellence Cluster Cardio Pulmonary System (ECCPS) and German Centre for Cardiovascular Research (DZHK) to RLM, as well as funding from the NIH (HL54737), Packard Foundation and Max Planck Society to DYRS.

# Additional information

## Competing interests

DYRS: Reviewing editor, *eLife*. The other authors declare that no competing interests exist.

## Funding

| Funder | Grant reference number | Author |
| --- | --- | --- |
| Human Frontier Science Program | LT001023/2012-L | Ryota L Matsuoka |
| Japan Society for the Promotion of Science | | Ryota L Matsuoka |
| Damon Runyon Cancer Research Foundation | DRG#2104-12 | Ryota L Matsuoka |
| Excellence Cluster Cardio-Pulmonary System | | Ryota L Matsuoka |

| German Centre for Cardiovascular Research | | Ryota L Matsuoka |
| Max-Planck-Gesellschaft | | Didier YR Stainier |
| National Institutes of Health | HL54737 | Didier YR Stainier |
| David and Lucile Packard Foundation | | Didier YR Stainier |

The funders had no role in study design, data collection and interpretation, or the decision to submit the work for publication.

## Author contributions

RLM, Conception and design, Acquisition of data, Analysis and interpretation of data, Drafting or revising the article, Contributed unpublished essential data or reagents; MM, Acquisition of data, Analysis and interpretation of data, Drafting or revising the article; AA, Acquisition of data, Analysis and interpretation of data; CSMH, ASG, NDL, WH, Drafting or revising the article, Contributed unpublished essential data or reagents; H-MM, HK, Acquisition of data, Drafting or revising the article; DYRS, Conception and design, Analysis and interpretation of data, Drafting or revising the article

## Author ORCIDs

Didier YR Stainier, http://orcid.org/0000-0002-0382-0026

## Ethics

Animal experimentation: This study was performed in strict accordance with institutional (UCSF and MPG) and national ethical and animal welfare guidelines. All of the animals were handled according to approved institutional animal care and protocol (Permission No. B2/1068).

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
