## [Decision Letter]

Thank you for submitting your article "Radial Glia Regulate Vascular Patterning Around the Developing Spinal Cord" for consideration by *eLife*. Your article has been reviewed by three peer reviewers, and the evaluation has been overseen by Marianne Bronner as the Senior Editor and Reviewing Editor. The following individuals involved in review of your submission have agreed to reveal their identity: Naoki Mochizuki (Reviewer #2); Yoshiko Takahashi (Reviewer #3).

The reviewers have discussed the reviews with one another and the Reviewing Editor has drafted this decision to help you prepare a revised submission.

Summary:

In this manuscript, Matsuoka et al. identify a previously unappreciated mechanism in which radial glia repress venous sprouting around the spinal cord. Employing a variety of genetic tools in zebrafish, they demonstrate that ablation of radial glia leads to excessive venous sprouting, that this excessive sprouting requires Vegfr/Vegfr2 signaling, and that the normal repression of excessive venous sprouts is dependent on sFlt1 function within venous cells. The experiments are well-designed, the data are nicely presented, and the interpretation of the results is clear-cut. The implication of sFlt1 as a repressor of vascular sprouting in this context is consistent with previous work on sFlt1 function in other settings. The particular significance of this work comes from its novel demonstration that radial glia play an important role in patterning the CNS vascular network.

Essential revisions:

Several major questions regarding the repression of venous sprouting by radial glia remain unanswered by the experiments presented here:

1) The authors conclude that radial glia have a positive influence on the levels of sFlt1 expression, and they imply that local contacts between radial glia and venous cells could mediate this influence. The mechanism through which radial glia promote sFlt1 expression remains unaddressed, and the authors even refrain from speculating regarding this mechanism in the Discussion section. What are possibilities for this mechanism, based on what is already known about the regulation of sFlt1? For example, have the authors considered whether a Semaphorin-Plexin interface could influence sFlt1 expression in this context, as has been previously shown in the context of the dorsal aorta (Zygmunt et al., 2011)? Identification of this (or any other) molecular link between radial glia and venous cells would enhance the value of the authors' model; discussion of possible molecular links would enhance the scholarly value of the manuscript.

2) It is clear that the effects of radial glia are highly selective: excess venous sprouting is abundant following radial glia ablation, but excess arterial sprouting is almost never observed. The impact of this work would be enhanced by a better understanding of the mechanism for this selectivity. In the Discussion, the authors suggest the possibility that mural cells might protect arterial cells from the influence of radial glia. Is it possible to begin to test this idea, using mutations or ablation alleles that disrupt mural cell coverage? For example, *pdgfrb^-/-^* larvae presumably lack pericytes, so if pericyte coverage accounts for the difference in arterial and venous sprouting, then it might be possible to observe sprouting from aISVs in *pdgfrb* mutants. However, Figure 3 clearly shows no branching from aISVs. Can the authors explain this apparent inconsistency and discuss why aISVs are not affected? The presence of mural cells does not account for the difference between aISVs and vISVs.

3) In Figure 6, it looks as if Flit1 is also positive in aISV. If so, why does radial glia ablation affect only vISV and not aISV?

---

## [Author Response]

[…]

Essential revisions:

Several major questions regarding the repression of venous sprouting by radial glia remain unanswered by the experiments presented here:

1) The authors conclude that radial glia have a positive influence on the levels of sFlt1 expression, and they imply that local contacts between radial glia and venous cells could mediate this influence. The mechanism through which radial glia promote sFlt1 expression remains unaddressed, and the authors even refrain from speculating regarding this mechanism in the Discussion section. What are possibilities for this mechanism, based on what is already known about the regulation of sFlt1? For example, have the authors considered whether a Semaphorin-Plexin interface could influence sFlt1 expression in this context, as has been previously shown in the context of the dorsal aorta (Zygmunt et al., 2011)? Identification of this (or any other) molecular link between radial glia and venous cells would enhance the value of the authors' model; discussion of possible molecular links would enhance the scholarly value of the manuscript.

We would first like to thank all the reviewers for noting the importance of our study and also their constructive comments. As suggested by the reviewers, we have now added a paragraph in the revised manuscript that discusses possible molecular links between radial glia and venous endothelial cells based on previous literatures on sFlt1 regulation.

2) It is clear that the effects of radial glia are highly selective: excess venous sprouting is abundant following radial glia ablation, but excess arterial sprouting is almost never observed. The impact of this work would be enhanced by a better understanding of the mechanism for this selectivity. In the Discussion, the authors suggest the possibility that mural cells might protect arterial cells from the influence of radial glia. Is it possible to begin to test this idea, using mutations or ablation alleles that disrupt mural cell coverage? For example, pdgfrb^-/-^ larvae presumably lack pericytes, so if pericyte coverage accounts for the difference in arterial and venous sprouting, then it might be possible to observe sprouting from aISVs in pdgfrb mutants. However, Figure 3 clearly shows no branching from aISVs. Can the authors explain this apparent inconsistency and discuss why aISVs are not affected? The presence of mural cells does not account for the difference between aISVs and vISVs.

We appreciate this suggestion. Regarding *pdgfrb^-/-^*larvae, these mutants indeed lack a majority of brain pericytes as previously shown by Ando K. et al., (Development: 143, 1328-39, 2016) as well as by our analysis using a *TgBAC(pdgfrb:citrine)^s984^* pericyte reporter line (data not shown). However, mural cells that cover ISVs appear to be only mildly affected in *pdgfrb^-/-^* larvae as shown in the mentioned article. We speculate that apparently decreased, yet incompletely eliminated, mural cells could still protect aISVs from over-sprouting after radial glia ablation, thus potentially explaining why only very few aISVs over-sprout in *pdgfrb^-/-^* larvae after radial glia ablation.

In order to more effectively ablate mural cells that cover ISVs, we performed additional experiments where mural cells were genetically ablated using *TgBAC(pdgfrb:gal4ff)^ncv24^* fish in combination with radial glia ablation. However, a vast majority of the fish in which both cell types were ablated by this method developed pericardial edema and an unhealthy appearance, making it more complicated and difficult to assess over-sprouting from aISVs and vISVs. We think that it may work slightly better to perform mural cell ablation in *flt1^bns29^* mutants rather than in radial glia-ablated fish. However, it may also lead to similar results as genetic ablation of mural cells using *TgBAC(pdgfrb:gal4ff)^ncv24^* fish basically caused pericardial edema and an unhealthy appearance. We think that inducing mural cell ablation using the *TgBAC(pdgfrb:gal4ff)^ncv24^* line is currently the best possible approach to test this hypothesis given the available tools. However, we are not sure whether this approach will completely eliminate all mural cells (pericytes and smooth muscle cells) that cover the ISVs; in addition, this experiment requires 4 different transgenes in *flt1^bns29^* mutants, namely *TgBAC(pdgfrb:gal4ff);Tg(UAS:mcherry-NTR);Tg(kdrl:EGFP);Tg(-0.8flt1:tdTomato)* and thus the generation and analyses of these animals will require substantial time, and may end up with an inconclusive answer for the reasons mentioned above. We thus would like to fully investigate the mechanisms and pursue this important question in future studies by generating and testing other possible hypotheses and tools. However, to clarify our hypothesis and rationale behind the potential contribution of perivascular cells to selective vISV responses in radial glia-ablated or *flt1^bns29^* mutant fish, we discuss these points more clearly in our revised manuscript by citing the mentioned article that reports the mildly affected mural cells in *pdgfrb^-/-^*larvae.

3) In Figure 6, it looks as if Flit1 is also positive in aISV. If so, why does radial glia ablation affect only vISV and not aISV?

We appreciate this comment. Closely related to the point 2), we speculate that there are additional mechanisms that protect aISVs from over-sprouting after radial glia ablation. As we described above, one hypothesis we have pursued is the aspect of distinct coverage of aISVs and vISVs by perivascular cells. However, the experiments we performed and plan to perform may not get us a conclusive answer due to the technical issues we have faced as mentioned above. In addition, there should be a lot more possibilities other than this hypothesis, and thus we would like to address this important issue fully in future studies.